# Soup-of-Experts: Pretraining Specialist Models via Parameters Averaging

**Pierre Ablin** [1]   **Angelos Katharopoulos** [1]   **Skyler Seto** [1]   **David Grangier** [1]

## Abstract

Machine learning models are routinely trained on a mixture of different data domains. Different domain weights yield very different downstream performances. We propose the Soup-of-Experts, a novel architecture that can instantiate a model at test time for any domain weights with minimal computational cost and without re-training the model. Our architecture consists of a bank of expert parameters, which are linearly combined to instantiate one model. We learn the linear combination coefficients as a function of the input domain weights. To train this architecture, we sample random domain weights, instantiate the corresponding model, and backprop through one batch of data sampled with these domain weights. We demonstrate how our approach obtains small specialized models on several language modeling tasks quickly. Soup-of-Experts are particularly appealing when one needs to ship many different specialist models quickly under a size constraint.

## 1. Introduction

Large Language Models (LLMs) work well on diverse tasks because they have many parameters and are trained on generalist datasets (Brown et al., 2020; Bommasani et al., 2021). However, they are costly to train and to serve, both in terms of memory and inference cost.

Specialist language models hold fewer parameters; they are, therefore, cheaper to store, send, and use at inference. However, they must give up the generality of LLMs and specialize in a few specific topics.

In cases where there is an abundance of specialization data, training a small model on those data yields a good specialist. However, in many settings, the specialization data is scarce: for instance, it may come from a narrow topic of interest

[1]Apple. Correspondence to: Pierre Ablin <p_ablin@apple.com>.

*Proceedings of the 42$^{nd}$ International Conference on Machine Learning*, Vancouver, Canada. PMLR 267, 2025. Copyright 2025 by the author(s).

or be a small company's internal document database. It is, therefore, impossible to train a good-quality specialist model on such data alone.

To obtain a small model that performs well on the specialization data, we leverage a large, generic pretraining dataset. That pre-training set contains data from several domains. A powerful method to obtain a good specialist model is importance sampling: it adjusts the mixture weights of the pretraining distribution to resemble the scarce specialist dataset. This method has been shown to outperform generic pre-training (Grangier et al., 2024b), but it has a major drawback: it requires pre-training a full model for each specialization dataset available. This makes training cost scale linearly with the number of specialized downstream tasks, which can be intractable as model size and data scales.

The goal of this paper is to answer the following question: *How can we leverage a large pre-training set to obtain specialized models that can be instantiated quickly when the specialization data is revealed?*

We formalize this question by considering the two phases of serving specialist models.

**Pretraining** We have multiple pre-training domains and use them to train a model. At this point, we do not know the specific data and are unaware of what specific tasks we will need to address later on.

**Specialization phase** We receive a specific dataset, and using the pre-trained model, we need to quickly instantiate a small model that works well on this specific dataset.

In Table 1, we summarize the different costs and constraints associated with these two phases and provide a qualitative review of the strengths and weaknesses of several strategies.

In this landscape of different models, we introduce the Soup-of-Experts, which is designed to be able to instantiate a small specialist model in a flash.

Our main idea is to learn to instantiate models with any mixture of domain weights by taking a linear combination of jointly optimized base models, called experts. We are inspired by the works of model merging (Wortsman et al., 2022; Arpit et al., 2022; Rame et al., 2022; 2023; 2024). The gist of model merging is that two model parameters $\Theta_A$ and $\Theta_B$ that are obtained by fine-tuning the same model

| Model | Spec. size | Pretrain. size | Pretrain. cost | Spec. Cost | Spec. Latency | Spec. Loss |
|---|---|---|---|---|---|---|
| Large generic model | Large | Large | Large | Null | Large | Small |
| Mixture of Experts | Large | Large | Small | Null | Small | Med |
| Small generic model | Small | Small | Small | Null | Small | Large |
| Domain Experts | Small | Large | Med | Small | Small | Med |
| CRISP | Small | Null | Null | Large | Small | Small |
| **Soup-of-Experts** | Small | Large | Small | Small | Small | Small |

*Table 1.* **The different quantities that matter during the phases of serving a specialized model.** Spec. size is the number of parameters in the specialized model. Pretrain. size is the total number of parameters of the pretrained model. Pretrain. cost is the cost of pretraining the model. Spec. cost is the cost to obtain a specialized model when the specialized data is made available. Spec. latency is the cost of performing inference with that model. Spec. loss is the loss on the specialized dataset. With these constraints in mind, we compare different models. The goal of this work is to propose the **best possible model under the constraint of having a small specialized model size**. Large generic model is a generalist model, with many parameters, that requires a long training. A mixture of Experts (Fedus et al., 2022a; Krajewski et al., 2024) is a small model with added parameters that marginally impact the latency. Since both the LLM and the MoE have many parameters, they are discarded from our study. Small generic model is one small model trained on a generalist distribution. Domain experts (Gross et al., 2017) train one small model per pre-training domain. CRISP (Grangier et al., 2024b) trains one model once the specialized data is available using a data mixture that imitates the specialized data distribution. Our proposed method, the Soup-of-Experts, trains one model with many parameters and can quickly instantiate a small model that is good on the specialized data. The results in this table are qualitative.

on different domains $A$ and $B$ can be merged by averaging, yielding a new model $\Theta^* = \frac{1}{2}(\Theta_A + \Theta_B)$, sometimes called a model *soup* (Wortsman et al., 2022), to obtain good performances on both datasets. An important lesson from model merging is that some models' parameters can be linearly combined and yield good models.

A caveat of model merging is that the merged models can only be fine-tuned versions of the same base model: for merging to work, the two models must not be too far apart in the parameters space. Our method, Soup-of-Experts, *pre-trains* multiple experts that can, by design, be linearly combined to yield a single specialized model. The linear coefficients of the combination are learned as a function of the pre-training domain weights. Figure 1 gives an overview of the architecture and the training pipeline.

**Paper overview** In Section 2, we explain the details of the Soup-of-Experts, its training pipeline, and how it can be used to instantiate a specialist model quickly. In Section 3, we demonstrate the promises of this approach in a standard language model pre-training setup, where we train small 110M models on Redpajamav2 (Weber et al., 2024) and specialize them on 16 domains from the Pile (Gao et al., 2020). We conduct several analyses to clarify the roles of model size, number of experts, training distribution, and specialized dataset size. Finally, in Section 4, we position our work within the literature.

## 2. Methods

Figure 1 gives an overview of the proposed architecture, its interplay with data, and its training pipeline. We first explain the training data setup.

---

**Algorithm 1** Sampling from $\mathrm{mix}(h) = \sum_{i=1}^{k} h_i D_i$

**Input:** Domains $D_1, \ldots, D_k$, domain weights $h \in \mathbb{R}^k$, batch size $B$
**for** $b = 1, \ldots, B$ **do**
    Sample an index $i \sim \mathrm{Categorical}(h)$
    Sample $x_b$ uniformly at random from domain $D_i$
**end for**
**Output:** Mini batch $[x_1, \ldots, x_B]$

---

### 2.1. Sampling from the pre-training set

The pre-training set is composed of $k$ domains $D_1, \ldots, D_k \subset \mathcal{X}$ where $\mathcal{X}$ is the sample space (in the case of LLMs, this is the space of text sequences). Each domain contains many samples, usually enough to train a model without repeating data or overfitting. We can query samples from each of these domains, therefore we can sample from a *weighted* mixture of domains: for some **domain weights** $h \in \mathbb{R}^k$, we define the sampling law $\mathrm{mix}(h) = \sum_{i=1}^{k} h_i D_i$ such that

$$P(x|\mathrm{mix}(h)) = \sum_{i=1}^{k} h_i P(x|D_i). \quad (1)$$

This law mixes the datasets $D_i$ with proportions $h_i$, where the domain weights $h$ are non-negative and sum to one. We can efficiently query samples from $\mathrm{mix}(h)$ for any domain weights $h$, by picking a domain $i$ at random following the categorical law induced by $h$, and then sampling an element at random from the corresponding domain $D_i$. The corresponding law is illustrated in Figure 2, and this strategy is described in Algorithm 1.

Classical generic pre-training uses fixed pre-training domain

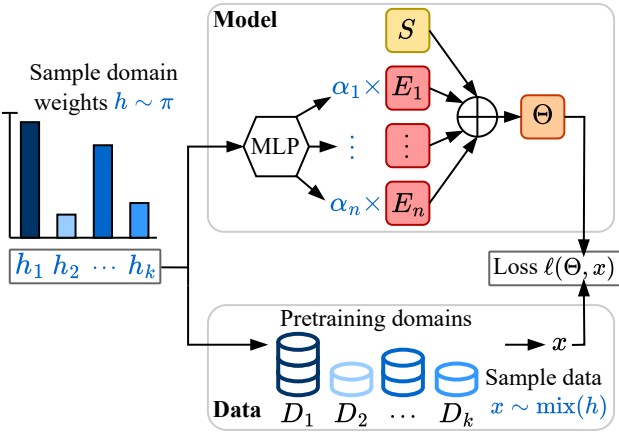

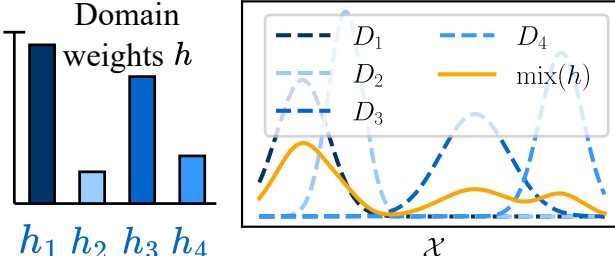

*Figure 2.* **Data mixture sampling** Given several pretraining domains $D_1, \ldots, D_k$, an domain weights $h_1, \ldots, h_k$, we can train a model on the mixture $\mathrm{mix}(h) = \sum_{i=1}^k h_i D_i$, using the sampling procedure described in Algorithm 1. Domain weights have a critical impact on the downstream performance.

*Figure 1.* **The Soup-of-Experts and its training pipeline.** The Soup-of-Experts consists of shared parameters $S$, $n$ experts parameters $E_1, \ldots, E_n$, and an MLP that acts as a routing mechanism. At each optimization step, we sample domain weights $h$ from a meta-distribution $\pi$. These domain weights have two purposes: they are passed through an MLP to give a vector of coefficients $\alpha$ that instantiates a **model** by combining the experts' weights, and they are used to sample a mini-batch of **data** following the domain weights law. We then backpropagate through the corresponding loss to update the parameters of the Soup-of-Experts.

weights $h_{\mathrm{generic}}$ which define a generic dataset

$$D_{\mathrm{generic}} = \mathrm{mix}(h_{\mathrm{generic}}).$$

These weights are defined to train large generalist models that perform well on average. Finding weights for a good average behaviour to train large models is difficult (Xie et al., 2023). For smaller models, even a good $\mathrm{mix}(h_{\mathrm{generic}})$ would yield a model far from strong specialists, i.e., giving a model good at everything but excellent at nothing.

## 2.2. Training with mixtures of pre-training domains

We let $\Theta \in \mathbb{R}^p$ the parameters of a model to be trained on the pre-training set. We define $\ell(\Theta; x)$ the loss function for a sample $x \in \mathcal{X}$ (the next token prediction loss in this paper, since we focus on language modeling). The standard LLM pretraining consists of running Adam (Kingma, 2014) to approximately minimize the **generic loss**

$$L_{\mathrm{generic}}(\Theta) = \mathbb{E}_{x \sim D_{\mathrm{generic}}} [\ell(\Theta; x)]$$

Alternatively, we can train a model on any given mixture with domain weights $h$ by running Adam on the loss

$$L(\Theta, h) = \mathbb{E}_{x \sim \mathrm{mix}(h)} [\ell(\Theta; x)]$$

Grangier et al. (2024b) showed that a powerful technique to obtain a good small model on a specific set $D_{\mathrm{spe}}$ is to i)

find domain weights $h_{\mathrm{spe}}$ such that $\mathrm{mix}(h_{\mathrm{spe}}) \simeq D_{\mathrm{spe}}$ and then ii) train the model by minimizing $L(\Theta, h_{\mathrm{spe}})$. This importance-sampling-based method called CRISP gives much better specialists than generic pre-training since it trains the model on a distribution that has lots of data and yet is close to the targeted specific distribution.

One caveat of this approach is that it requires retraining a model from scratch anytime one wants to obtain a specialized model. While this cost might be justified in some critical applications, we study alternative avenues to obtain specialized models at a much smaller cost: this is the purpose of the new architecture that we propose in this paper, the Soup-of-Experts.

### 2.3. Soup-of-Experts

The goal of the Soup-of-Experts is to amortize the training of models on multiple different domain weights. It defines a method that, given training domain weights $h \in \mathbb{R}^k$, quickly instantiates a model $\Theta$ that depends on those domain weights and that yields a low loss $L(\Theta, h)$.

To do so, we enhance the base model with $n$ *experts* $E_1, \ldots, E_n \in \mathbb{R}^p$, which for ease of notation we stack into a matrix $\mathbf{E} = [E_1, \ldots, E_n] \in \mathbb{R}^{n \times p}$. We linearly combine the weights with shared parameters $S \in \mathbb{R}^p$. For a given set of expert coefficients $\alpha \in \mathbb{R}^n$, we instantiate a small model as

$$\Theta = \mathrm{Combine}(S, \mathbf{E}, \alpha) = S + \sum_{j=1}^n \alpha_j E_j. \quad (2)$$

Our main idea is to learn coefficients $\alpha$ as a function of the domain weights $h$. To be more precise, we want to learn parameters $S$, $\mathbf{E}$, and a function $\phi : \mathbb{R}^k \to \mathbb{R}^n$ such that, for any domain weights $h \in \mathbb{R}^k$, the instantiated model $\Theta = \mathrm{Combine}(S, \mathbf{E}, \phi(h))$ performs well on the dataset $\mathrm{mix}(h)$, i.e., leads to a low loss $L(\Theta, h)$. In practice, we use a two-layer MLP for $\phi$, parameterized by parameters $\omega$, denoted as $\phi_\omega$. Although one can think of many different

**Algorithm 2** Pre-training loop for a Soup-of-Experts to minimize the loss function $\mathcal{L}(S, \mathbf{E}, \omega)$ in Equation 3.

---

**Input:** Initial parameters $Z = (S, \mathbf{E}, \omega)$, domain weights sampling law $\pi$, domains $D_1, \ldots, D_k$, optimizer `optim`, optimizer state $s$.

**for** $t = 0, \ldots, T - 1$ **do**

    Sample a random domain weights $h \sim \pi$

    Sample $x \sim \mathrm{mix}(h)$ using Algorithm 1

    Compute gradients $g$ of the parameters by backpropagation through the loss $\ell(\mathrm{Combine}(S, \mathbf{E}, \phi_\omega(h)), x)$

    Update parameters and optimizer state: $Z, s \leftarrow$ `optim`$(g, Z, s)$

**end for**

**Output:** Learned parameters $S, \mathbf{E}, \omega$

---

ways to define a mapping from domain weights to model weights, we chose the parameterization in Equation 2 as it allows us to easily scale the number of total parameters (by increasing the number of experts $n$), and we know from the model merging literature that, perhaps surprisingly, different model parameters can be linearly combined to yield one good model(Wortsman et al., 2022).

The Soup-of-Experts is an *asymmetrical* model in the sense that it has many trained parameters (the base model and the added expert's parameters), but it instantiates smaller, stand-alone smaller models for inference.

We now explain how to leverage a large pre-training set in order to train a Soup-of-Experts.

### 2.4. Training Soups of Experts with meta-distributions

In order to train the Soup-of-Experts to achieve good performance on a diversity of domain weights, we use a **meta-distribution** $\pi$, that is, a sampling law over domain weights.

For instance, one can define $\pi$ as the uniform distribution over histograms, by sampling $\tilde{h}_1, \ldots, \tilde{h}_k$ i.i.d. uniformly in $[0, 1]$ and defining the domain weights as $h_i = \tilde{h}_i / \sum \tilde{h}_j$.

We then train the Soup-of-Experts by minimizing the average error of the model over this meta-distribution, which is the objective function

$$\mathcal{L}(S, \mathbf{E}, \omega) = \mathbb{E}_{h \sim \pi} \left[ L(\mathrm{Combine}(S, \mathbf{E}, \phi_\omega(h)), h) \right] \quad (3)$$

We minimize this function using Adam, where at each step, we sample domain weights $h \sim \pi$, instantiate the corresponding model, sample a mini-batch from $\mathrm{mix}(h)$, and do an optimization step on $\ell(\mathrm{Combine}(S, \mathbf{E}, \phi(h)), x)$. The full algorithm is described in Algorithm 2, and Figure 1 illustrates this training pipeline.

The choice of meta-distribution $\pi$ has a critical role on the Soup-of-Experts obtained after training. Ideally, it should reflect the distribution of specific tasks that one wishes to address during the specialization phase. In our experiments, we favor sparse domain weights and use meta-distributions $\pi$ that first sample $s \ll k$ domains and then take uniform random domain weights over these $s$ domains.

### 2.5. Computationnal cost

The Soup-of-Experts leads to some computational overhead compared to standard pre-training, which we explicit here. We compare training a generic model of size $d$ with standard pre-training and training a Soup-of-Experts with $n$ experts of size $d$. The routing MLP is small compared to the model size; hence, we omit the cost of training it in the subsequent analysis. We let $b$ be the mini-batch size.

The training loop of models consists of A) computing gradients and B) updating the parameters with Adam. For the generic model, the cost of A) scales roughly as $db$ (Kaplan et al., 2020), we let $C$ be the proportionality constant. This computes $\nabla_\Theta \ell(\Theta, x)$, with a cost $Cdb$. For the Soup-of-Experts, we need to compute the gradients with respect to the experts $E_i$ and shared parameters $S$. Let $\Theta = S + \sum \alpha_i E_i$ the merged model. Then, the chain rule gives the gradients $\nabla_S L = \nabla_\Theta \ell(\Theta, x)$ and $\nabla_{E_i} L = \alpha_i \nabla_\Theta \ell(\Theta, x)$. In other words, the gradients w.r.t. the experts are simply obtained by rescaling the gradient wrt the merged parameters, which gives $n \cdot b$ floating point operations. Hence, the cost of A) is $C \cdot d \cdot b + n \cdot b$ for the Soup-of-Experts. Crucially, the cost of backprop through the Soup-of-Experts is only mildly affected by n, as the gradients of the experts are obtained trivially from those of the merged model.

For B), the generic pretraining needs to update the two Adam EMA parameters, and then update the parameters. In total, the flops count is 14d. For the Soup-of-Experts, we need to use Adam for all the parameters, so the cost of B) is $14n \cdot d$.

Overall, one training iteration for generic pretraining costs $C_{\mathrm{Gen}} = Cdb + 14d$, while it is $C_{\mathrm{SoE}} = Cdb + 15nd$ for the SoE. Discarding the cost of Adam for the generic pretraining, the relative increase is $C_{\mathrm{SoE}}/C_{\mathrm{Gen}} = 1 + \frac{15n}{Cb}$. Hence, we see that it all depends on whether $n \gg Cb/15$; the value of the constant $C$ depends on the architecture and other factors. In a large batch size $b$ setting, the added cost of using a Soup-of-Experts is negligible.

The cost of training a Soup-of-Experts of size $d$ with $n$ experts is close to that of training a single model of size $d$, and therefore it is *much lower* than that of training a single large $n \times d$ model. For instance, in the subsequent experiments, we train Soup-of-Experts with 128 experts of size $110M$ in about a day using a grid of 8GPUs. Training a model of size $128 \times 110M = 13B$ would take orders of

**Algorithm 3** (Grangier et al., 2024b) Estimating specialist domain weights that are good for a specialized dataset $D_{\text{spe}}$

> **Input:** Specialist dataset $D_{\text{spe}}$, embedded domain centroids $c_1, \ldots, c_k$.
> **Init:** $h_1, \ldots, h_k = 0$.
> **for** $x$ in $D_{\text{spe}}$ **do**
>     Find closest centroid: $i = \arg\min \|\texttt{Bert}(x) - c_i\|$
>     Increment $h_i = h_i + 1/\#D_{\text{spe}}$
> **end for**
> **Output:** Domain weights $h_1, \ldots, h_k$

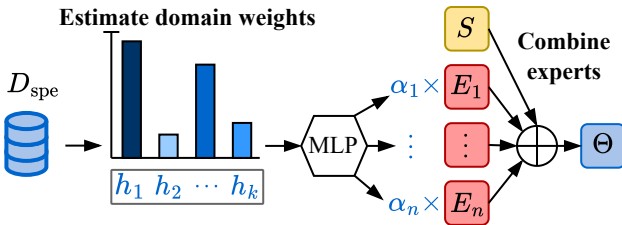

*Figure 3.* **Quickly instantiating a small model from a pretrained Soup-of-Experts** Given a specialist dataset with a few samples, we compute the domain weights using Algorithm 3. The domain weights are then passed through the Soup-of-Experts' MLP to get the coefficients $\alpha$ that are then used to merge the experts. This process is quick since the MLP is small, and it requires no training.

magnitude more compute, typically weeks with 128GPUs (see (Touvron et al., 2023)).

### 2.6. Parameter efficient representation with low-rank expert

Thus far, the experts have the same size as the original model. In order to diminish the total number of parameters, we can use a low-rank representation for the experts in the spirit of Lora (Hu et al., 2021): in each expert $E_j$, each square matrix $W \in \mathbb{R}^{a \times b}$ is materialized as $W = AB^T$ where $A \in \mathbb{R}^{a \times r}$ and $B \in \mathbb{R}^{b \times r}$, where $r \ll a, b$ is the rank. Interestingly, even though each expert's matrices are low rank, the instantiation $\sum \alpha_j E_j$ has rank up to $n \times r$, which might be full rank if we have enough experts. Although promising, we report in our experiments that this method is less parameter-efficient than using fewer dense experts. Still, in heavily resource-constrained settings, low-rank experts can benefit over generic model.

### 2.7. Instantiating a Soup-of-Experts: Specialization in a flash

After pre-training, the Soup-of-Experts has the flexibility to quickly provide a model that is good for any data distribution domain weights $h$, simply by forming the parameters $\Theta = \text{Combine}(S, \mathbf{E}, \phi(h))$. This instantiation only requires a forward pass through a small MLP, and merging $n$ parameters; it does not require any training.

We describe two ways to specialize a Soup-of-Experts into a model that performs well on a target specific dataset $D_{\text{spe}}$. The fastest way is to obtain domain weights $h_{\text{spe}}$ from $D_{\text{spe}}$ so that $D_{\text{spe}} \simeq \text{mix}(h_{\text{spe}})$. To do so, we use the nearest-neighbor method of (Grangier et al., 2024b), which is described in Algorithm 3 for completeness. We then instantiate the parameters $\text{Combine}(S, \mathbf{E}, \phi(h_{\text{spe}}))$. This method is summarized in Figure 3. Since it is simple and fast, this is the method we use in all our experiments.

A better method, which is also more expensive, is to learn a coefficient vector $\alpha$ with gradient descent, by minimizing the function of $\alpha$ only $\psi(\alpha) =$

$\mathbb{E}_{x \sim D_{\text{spe}}} [\ell(\text{Combine}(S, \mathbf{E}, \alpha), x)]$. Since $\alpha$ is low dimensional (in practice we never use more than $n = 128$ experts), this minimization is quick, and unless $D_{\text{spe}}$ has very few samples, there is no risk of overfitting. However, this method requires to backpropagate through the network, which is more costly than the previous method.

As with any other model, the instantiated specialist model can then be fine-tuned on the specialization data to increase its performance if the computational budget allows it.

## 3. Experiments

We first detail the experimental setup: datasets, models, metrics, and hyperparameters.

**Pretraining domains** We pre-train language model on Redpajama2 (Weber et al., 2024), a widely used curated web-crawl dataset. We obtain the pre-training domains $D_1, \ldots, D_k$ with the same clustering method as Grangier et al. (2024b): we embed each document using sentence-bert (Devlin, 2018), and then use the k-means algorithm on these embeddings to split the dataset into $k$ pre-training domains. We use a hierarchical k-means, where we first cluster the dataset into $k = 64$ domains and then cluster each of these domains into 64 smaller domains, yielding in total $k = 4096$ domains. We also collect the $k$ corresponding centroids $c_1, \ldots, c_k$ in the embedding space, in order to use Algorithm 3 to obtain specialist domain weights.

**Specialization domains** We consider 16 datasets from the PILE (Gao et al., 2020) as target specialization sets: arxiv, dm_mathematics, enron emails, europarl, freelaw, github, hackernews, nih exporter, openwebtext, pg19, phil papers, pubmed, stackexchange, ubuntu, uspto, and wikipedia.

For each of these datasets, we compute the corresponding specialist domain weights using Algorithm 3.

We evaluate different methods on each of the specialization

datasets individually, and we report averaged losses over these domains. We defer individual domain results to the appendix.

We highlight that these specialist domains and specialist domain weights are never used or seen during the pre-training phase for all methods except for CRISP.

**Models** We consider standard GPT-2 type transformer architectures, which we train with the next-token-prediction loss. Apart from a scaling experiment, we consider a base model size of $110M$ parameters. Architecture and training hyperparameters are specified in Appendix A.

**Metrics** In this work, we measure the ability of a model on a specialization dataset with its next-token prediction loss on that domain: we focus solely on language modeling. This loss predicts well the downstream performance of models with more complex metrics like reasoning, question-answering or translation ability (Gonen et al., 2022; Du et al., 2024; Gadre et al., 2024).

**Training hyperparameters for the Soup-of-Experts** Unless specified otherwise, we train the Soup-of-Experts with $n = 128$ experts. With a base model size of $110M$, these Soup-of-Experts therefore hold a total of $(128 + 1) \times 110M = 14B$ parameters, that can be linearly combined into small $110M$ models. Apart from the corresponding analysis, we use a meta-distribution $\pi$ with a support size of $s = 4$ (see Section 2.4).

**Infrastructure** We train each model on 8 A100 GPUs.

### 3.1. Baselines

All the methods we compare in this work instantiate, at specialization time, a model with the same architecture and number of parameters. As explained in the introduction (Table 1), we consider the following models:

**Generic Pretraining** We train one generic model on the standard pre-training distribution. At specialization time, the model stays the same and is evaluated on the specialization set.

**Domain experts** (Gross et al., 2017) We train one model on each pretraining domain $D_i$. At specialization time, we select the model that yields the smallest loss on the specialization set. This technique does not scale with the number of domains. We only train $k = 64$ domain experts, as it would be infeasible to train 4096 with our budget.

**CRISP** (Grangier et al., 2024b) We train one model per specialization set on the mixture $\mathrm{mix}(h_{\mathrm{spe}})$. At specialization time, we use the corresponding model. This method does not scale with the number of specialization domains; it requires one pre-training run per specialization domain.

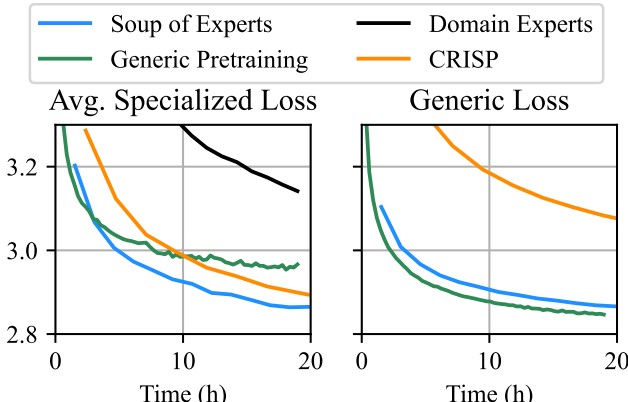

*Figure 4.* **Training curves of the different methods.** The average specialized loss is the average of the loss of the models over 16 domains from the Pile. The generic loss is the loss of the models on the standard pre-training distribution of RedPajamav2. The x-axis is the training time. This number is roughly proportional to number of tokens processed, since in this setting, the cost of instantiating the Soup-of-Experts is small in front of that of backpropagating through the network. The domain experts and CRISP have to train many models, so they are not competitive in this setup. The Soup-of-Experts performs almost similarly to generic pre-training on the generic loss, which means that it holds the general knowledge in the pre-training set, while CRISP and Domain Experts are not good generalists (Domain Experts are even out of the figure limits on the right figure). The Soup-of-Experts gives the best specialists, as seen on the left figure.

### 3.2. Main results

We report the training curves on the pre-training set as well as the average loss on the specialization domains in Figure 4. The specialized loss is obtained by computing the loss on each specialization domain for the corresponding specialist domain; each domain uses a different model (except for the generic pretraining method, which uses the same model for each specialization set).

The x-axis corresponds time, which in this case is close to being proportionnal of the computational cost required to train the model (indeed, the cost of instantiating the experts is small in front of that of backpropagating through the network, see Section 2.5; we get a throughput with the SoE that is 77% of that of the generic pretraining).

For the Soup-of-Experts and the generic pre-trained models, the training time is unambiguous. For the two other baselines, which train multiple models, we report the total training time taken by all the models.

We observe that the Soup-of-Experts achieves the best performance among all methods on the specialized domains, and is only slightly worse than generic pre-training on the pre-training loss (while generic pre-training explicitly minimizes this loss).

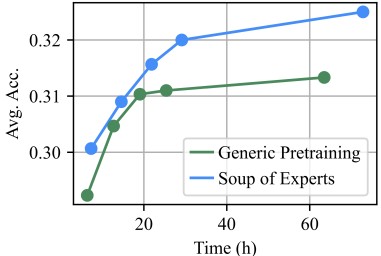

Figure 5. **Soup-of-Experts improve downstream tasks.** We train Soup-of-Experts with 335M parameters and 32 experts. We report the average accuracy of these models on MMLU, ARC-Easy and ARC-Challenge as a function of the training time using 8GPUs. The Soup-of-Experts is always more efficient, and training it for a long time yields a higher plateau.

The Soup-of-Experts and the generic pretraining are the only scalable methods with respect to the number of pretraining domains and number of specialization domains. Indeed, we consider 16 specialization domains here. Had we considered more domains, the CRISP method would have taken more and more pre-training time. Similarly, increasing the number of domains would increase the computational cost of the domain experts method a lot.

**Improvements on downstream tasks** In order to verify that the improvements in terms of loss transfer to downstream tasks, we train a larger Soup-of-Experts (335M parameters) with fewer experts ($n = 32$). We compare it to generic pretraining. We then evaluate it on question answering tasks: MMLU, ARC-Easy and ARC-Challenge. To instantiate a Soup-of-Experts on these tasks, we take a held-out part of these datasets, and use the questions as the specific set, for which we compute the histogram using Algorithm 3. We report the average accuracy on those tasks in Figure 5

**Complementarity to fine tuning** For each of the pile domains, we instantiate the corresponding Soup-of-Experts. We then fine-tune this model and the baseline model with different numbers of available fine-tuning tokens. We report the validation losses in Figure 6, as well as the number of tokens the generic pretraining method needs to use to recover a performance similar to that of the Soup-of-Experts.

### 3.3. Analysis

We quantify the impact of several hyper-parameters on the behavior of the Soup-of-Experts.

**Model scale** We train generic pretrained models and Soup-of-Experts with different instantiated model sizes. We report those results in Figure 7, left.

**Domain weights estimation** The specialized domain weights, computed with Algorithm 3, are estimated as frequencies. When we have little data available in the spe-

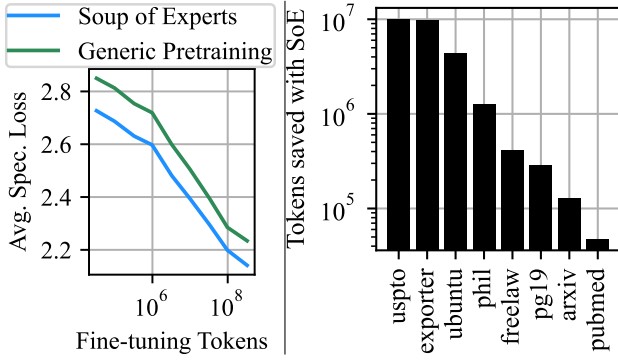

Figure 6. **The gains of Soup-of-Experts during pretraining are maintained during fine-tuning and sometimes lead to large savings.** On each of the 16 domains from the PILE, we fine-tune the corresponding instantiated Soup-of-Experts and generic model, with a limited number of fine-tuning tokens. We stop fine-tuning at the point where validation loss starts increasing. **Left:** Average loss over domains. We see that the Soup-of-Experts maintains its advantage regardless of the number of available fine-tuning tokens. **Right:** The number of fine-tuning tokens one needs to fine-tune the generic model to reach the same validation loss as the *base*, not fine-tuned, Soup-of-Experts. For example, on uspto, one needs $10M$ tokens to fine-tune the generic model and reach the same loss as the Soup-of-Experts instantiated on uspto out of the box after pre-training.

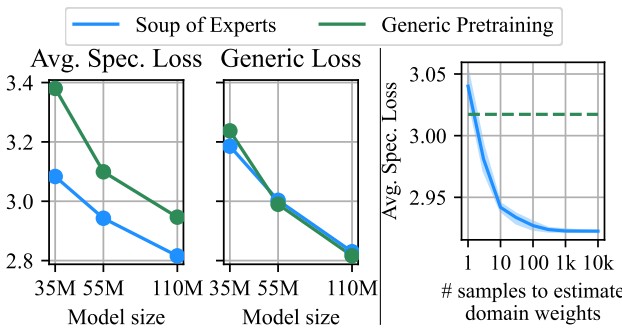

Figure 7. **Left: Impact of model scale** We train Soup-of-Experts and generic models with different instantiated model sizes. We observe that Soup-of-Experts maintain their advantage across the three scales considered here. **Right: Impact of the number of samples in the specialization set** In order to instantiate a specialist model from a Soup-of-Experts, we need to estimate the domain weights corresponding to the specialized set, as explained in Figure 3. We investigate the impact of the scarcity of data in the specialist set on the estimation of the domain weights, which then impacts the instantiated Soup-of-Experts. We see that on average, only three samples are enough to recover a model that is as good as the generic pretrained model, and 100-1000 samples are enough to instantiate the optimal Soup-of-Experts.

cialization set, the estimated domain weights become noisy.

We study how data scarcity impacts the performance of the Soup-of-Experts, using a limited number of samples as input to Algorithm 3. We report the results in Figure 7, right.

**Meta-distribution sampling law** We study the impact of the sampling law $\pi$ on the Soup-of-Experts performance. We train several Soup-of-Experts with different support sizes $s$, as explained in Section 2.4. We report the impact of $s$ on the loss on the specialist datasets, the generic dataset, and on random sparse domains in Figure 8.

**Low rank experts** As discussed in Section 2.6, we investigate the use of low-rank experts. The advantage of this method is that, at a fixed total number of parameters count, we can increase the number of experts and hence the ability of the Soup-of-Experts to have a fine-grained representation of the diversity of training domains. Sadly, as we report in Figure 9, this method is less parameter efficient than having dense experts. We posit that this is an optimization issue. Indeed, a Soup-of-Experts with experts with a high rank is able, in principle, to learn weights that are very similar to the dense one. Yet, in practice, it is harder to train.

**Comparison to model merging** We compare the Soup-of-Experts with standard model merging. To do so, we train both a Soup-of-Experts with k=64 domains and a generic model for 64000 iterations. Then, we fine-tune the generic model for T steps on each of the $k$ domains, yielding models $\theta_1, \ldots, \theta_k$ which are specialists for each domain. We then take $k' < k$ domains among the $k$ at random, and either a) instantiate the SoE with weights uniform over the $k'$ domains and $0$ over the other domains, or b) merge the specialized models $\theta_i$ where $i$ covers the $k'$ domains. We use a linear combination of models to merge. We then report the average loss of the corresponding model a) or b) on the $k'$ chosen domains. For the model merging, we always pick the number of fine-tuning steps T in [1000, 2000, ..., 10000] that yields the smallest loss for each individual experiment.

When there is only one domain, fine-tuning is the best method, since it gives a specialist. If we want good models on more than 4 domains, Soup-of-Experts becomes advantageous. Importantly, the SoE did not have to be trained on each domain individually: in total, it has done 64000 steps, while the model merging required $64000 + 64 \times 10000$ steps.

## 4. Related Work

The simultaneous growth of training set sizes, computational budgets and parameter count has yield large language models (LLMs) strong at addressing a variety of tasks (Brown et al., 2020; Jiang et al., 2023; Dubey et al., 2024).

The effectiveness of these generalist models comes at a high inference and training cost. To improve inference effectiveness, significant effort has been invested in knowledge distillation (Gu et al., 2024; Riviere et al., 2024) and model compression (Li et al., 2024; Wan et al., 2024). Specialization and sparsification are also important, complementary strategies for efficient inference.

Specialization trades generality for inference efficiency: a small model trained on data close to the targeted domain can be strong on this domain. Since many tasks only provide little in-domain data, fine-tuning a generalist model for a few steps is a common strategy. Data selection is a complementary approach that resamples the pretraining set to emphasize the most relevant parts for the target domain, allowing a specialist model to be trained from scratch. Data selection based on gradient alignment (Fan et al., 2023; 2024; Grangier et al., 2024a; Wang et al., 2024) and importance sampling (Grangier et al., 2024b) are the main strategies for task-aware pretraining.

Sparsification improves inference efficiency with mixture of experts (MoE). These models avoid using all parameters for all inputs. They jointly learn expert modules providing specialized weights for different data slices, and routing models deciding which weights to use (Shazeer et al., 2017; Fedus et al., 2022b; Jiang et al., 2024; Dai et al., 2024; Abnar et al., 2025). MoEs focus on the computational cost of inference but not on its memory cost (Pan et al., 2024): their routing decision are performed once the input is available, which means that they still require access to all the weights in memory. Model pruning (Xia et al., 2022; 2023; Ma et al., 2023) focuses on task-dependent sparsification as a fine tuning step, which allows memory saving. However, such pruning strategies are difficult since their pretraining has no incentive to discover sparse structures.

This work proposes an alternative. Like an MoE we train a large number of parameters but still achieve efficient inference. Unlike MoEs, we specialize the model to a given test domain and provide a small, stand-alone model. Unlike fine-tuning or task-aware pretraining, the specialized model does not require knowing the test domain at training time: specialization is not the result of optimization as the specialized parameters results from a closed form merging of the pretrained parameters. An interesting future avenue of research is to combine MoEs and Soup-of-Experts, where the base architecture in the Soup-of-Experts is itself a MoE. It would increase the performance of the Soup-of-Experts without sacrificing its latency.

This work takes inspiration from litterature on the merging of fine-tuned models, aka task-arithmetic. This litterature observes that models fine-tuned from a common ancestor can be linearly combined without retraining (Wortsman et al., 2022; Ilharco et al., 2022; Huang et al., 2023; Ortiz-Jimenez et al., 2023; Tam et al., 2024). It has also been proposed to further fine-tune merge models (Choshen et al., 2022; Rame et al., 2023) Our work extends this merging strategy beyond

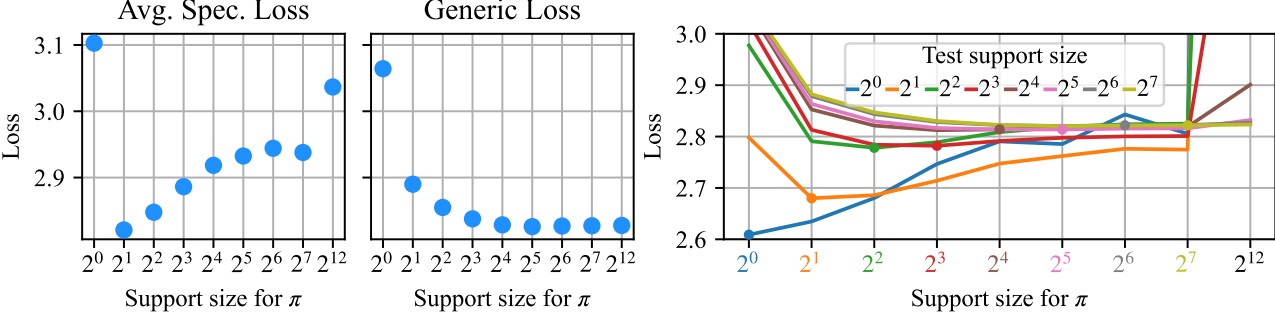

Figure 8. **Role of the support size for the meta-distribution $\pi$.** The training meta-distribution $\pi$ draws random domain weights by first sampling $s$ random domains, and then takes domain weights uniformly at random on those $s$ domains. We investigate the impact of $s$, the support size. A small value of $s$ means that the Soup-of-Experts does not see much interaction between domains, while a large $s$ means that the Soup-of-Experts rarely sees sparse domains, which are critical for specialized downstream tasks. **Left and middle**: average specialized and generic loss when varying the support size of the meta-distribution. As expected, taking a support size of 1 is bad, since the Soup-of-Experts cannot learn links between domains. The generic loss gets better as the support size increases, which is expected since the generic distribution is spread across all domains. The specialized loss is best for $s = 2$, which is explained by the sparsity of the specialized set domain weights. **Right**: average loss on random mixtures of fixed support size. We see that the training conditions are reflected in testing: the best Soup-of-Experts to test on domains of support size $s$ is the Soup-of-Experts trained with $s$ domains.

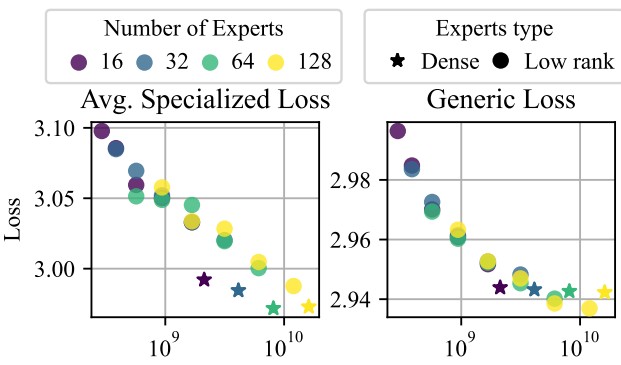

Figure 9. **Low-rank experts** We investigate the possibility to use low-rank experts as detailed in Section 2.6. Low rank experts allow increasing the number of experts at a fixed total number of parameters, by diminishing the rank of experts. We train multiple Soup-of-Experts with low rank experts, varying the number of experts in $\{16, 32, 64, 128\}$ and rank in $\{32, 64, 128, 256, 512\}$, yielding models that have between 300M and 12B parameters in total between the experts. We observe that, **at a fixed parameters count, the number of experts for low rank experts does not matter**. We also train standard Soup-of-Experts with dense experts, and we see a different trend. We find that **low-rank experts are worse than dense experts.**

fine-tuning and incorporates it into the pretraining phase.

(Dimitriadis et al., 2023) also propose an architecture that dynamically mixes weights according to the input task, but there are key differences with our work. First, they consider tasks that share the data but differ in their losses, while we consider tasks that have different data distributions with the same loss. Second, their framework does not allow having a

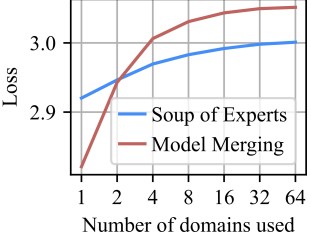

Figure 10. **Comparison to model merging** Soup-of-Experts allows to mix more efficiently a high number of domains that model merging, while requiring no fine-tuning from a base model.

different number of experts than domains, while we use a MLP to map histograms to experts. Finally, we introduce shared parameters in addition to the experts, which allows us to amortize between tasks. We discuss the differences with this work in more detail in Appendix B.

## Conclusion

We have introduced a novel asymmetrical architecture, the Soup-of-Experts. It holds a large set of expert parameters that encodes a family of small, stand-alone models obtained by linear combination of the parameters. We propose a learning algorithm so that the coefficients of the linear projection are a function of the domain weights from which the input is sampled. A pre-trained Soup-of-Experts can, therefore, instantiate instantly a model tailored to any mixture of domain weights. We demonstrated the benefits of this approach on standard datasets, even when these datasets and the corresponding domain weights are unavailable when the soup is trained.

## Impact Statement

This paper presents work whose goal is to advance the field of Machine Learning. There are many potential societal consequences of our work, none of which we feel must be specifically highlighted here.

## Acknowledgment

The authors are indebted to Alaaeldin El Nouby and Marco Cuturi for their insightful comments and suggestions. The authors heavily relied on a codebase that was kick-started by Awni Hannun. The authors thank Arno Blaas for his careful proofreading. Finally, the authors warmly thank Sophie Lepers, Arnaud Lepers, and Nicolas Berkouk for their judicious suggestions in the design and organization of the figures.

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

## A. Training hyper-parameters

| Model size | Vocab. size | Embedding size | Hidden MLP dim | layers | num heads |
|:---:|:---:|:---:|:---:|:---:|:---:|
| **110M** | 32K | 768 | 3072 | 12 | 12 |
| 55M | 32K | 512 | 2048 | 12 | 8 |
| 35M | 32K | 512 | 2048 | 6 | 8 |

*Table 2.* Model architectures. We use GPT-2 style transformers. All experiments except the scaling one use the 110M model. We use shared embedding matrices for input and output.

Table 2 details the model architectures used in the experiments.

| Hyperparameter | Value |
|:---:|:---:|
| Batch size | 128 |
| Sequence length | 1024 |
| Learning rate | 3e-4 for generic pre-training, domain experts and CRISP; 1e-4 for SoEs |
| Warmup steps | 2000 |
| Adam $\beta_1$ | 0.9 |
| Adam $\beta_2$ | 0.999 |
| Gradient clipping | 0.1 |

*Table 3.* Training hyperparameters.

We report the training hyperparameters in Table 3. After a search of learning rate in $\{$1e-4, 3e-4, 1e-3$\}$, we found that the best learning rate for the Soup-of-Experts was 1e-4, while it was 3e-4 for the other models.

We use different number of iterations for the different analyses: for the main experiments (Figure 4, Figure 6), we train the Soup-of-Experts and the generic pre-training model for $1024K$ iterations (134B tokens), while we train the domain experts and CRISP for $128K$ iterations (17B tokens), since we need to train multiple versions of those models.

For the model size analysis (Figure 7), we train for $1024K$ iterations (134B tokens). For the support size experiment (Figure 8), and for the low-rank experts experiment (Figure 9), we train for $128K$ iterations (134B tokens).

## B. Comparison with (Dimitriadis et al., 2023)

Using the notation of our paper, it is possible to reframe the model of (Dimitriadis et al., 2023) in the following way.

They consider a distribution of tasks $L_1, \ldots, L_k$ that are loss functions from $\mathbb{R}^p \times \mathcal{X}$ to $\mathbb{R}$. They consider a matrix of task interaction $W \in \mathbb{R}^{k \times n}$, and $n$ experts $E_1, \ldots, E_n$. Given a meta-distribution $\pi$ over the simplex of dimension $n$, they consider the following loss function, which should be optimized with respect to the experts parameters:

$$\tilde{L}(E) = \mathbb{E}_{x \sim \mathcal{D}} \left[ \mathbb{E}_{h \sim \pi} \left[ \sum_{i=1}^{k} (Wh)_i L_i(\sum h_i E_i; x) \right] \right]. \tag{4}$$

where $\mathcal{D}$ is the distribution of the inputs $x$. In our view, the best way to make this formulation as close as the one presented in this paper is to consider that $\mathcal{D}$ is the product space of the pretraining domains $\mathcal{D} = D_1 \times \cdots \times D_k$, that has elements $(x_1, \ldots, x_k)$ and to consider the loss functions $L_i(\theta; (x_1, \ldots, x_k)) = \ell(\theta; x_i)$, and to take $W = I_k$. Then, Equation 4 becomes

$$\tilde{L}(E) = \mathbb{E}_{h \sim \pi} \left[ \sum_{i=1}^{k} h_i \mathbb{E}_{x \sim D_i} \ell(\sum h_i E_i; x) \right] \tag{5}$$

$$= \mathbb{E}_{h \sim \pi} \left[ \sum_{i=1}^{k} L(\sum h_i E_i; h) \right] \tag{6}$$

This formulation highlights the key differences with our work:

- There are no shared parameters in the experts, while we have shared parameters in the experts that allow to amortize computations

- The input domain weights are used directly to mix the experts, while we use a small MLP to map the domain weights to the experts. This allows us to have a different number of experts than domains.

Finally, in the original formulation of Equation 4, the expectation is taken over domains, and the algorithm proposed by Dimitriadis et al. (2023) minimizes the loss over this expectation. In our setting where $\mathcal{D}$ is a product space, it means that the algorithm of Dimitriadis et al. (2023) needs to query samples from *each domain* in order to do one step of optimization, while our training loop only queries samples from one mixture of domains at a time.

# C. Detailed per-specific-domain results

We report the detailed results on the 16 PILE domains in Figure 11,Figure 12, Figure 13, and Figure 14.

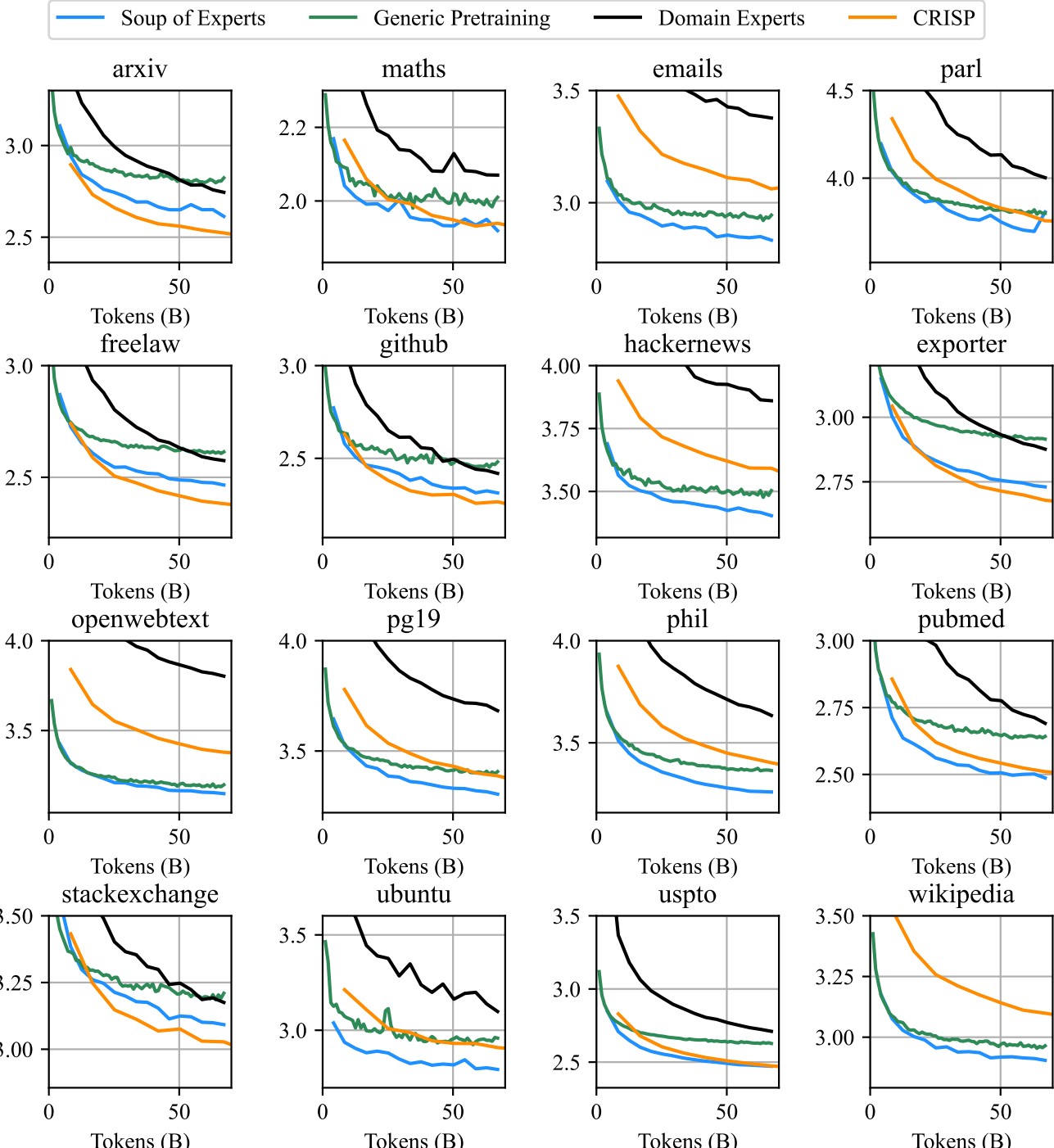

*Figure 11.* **Training curves of the different methods.** Detailed results from Figure 4 on the 16 PILE domains.

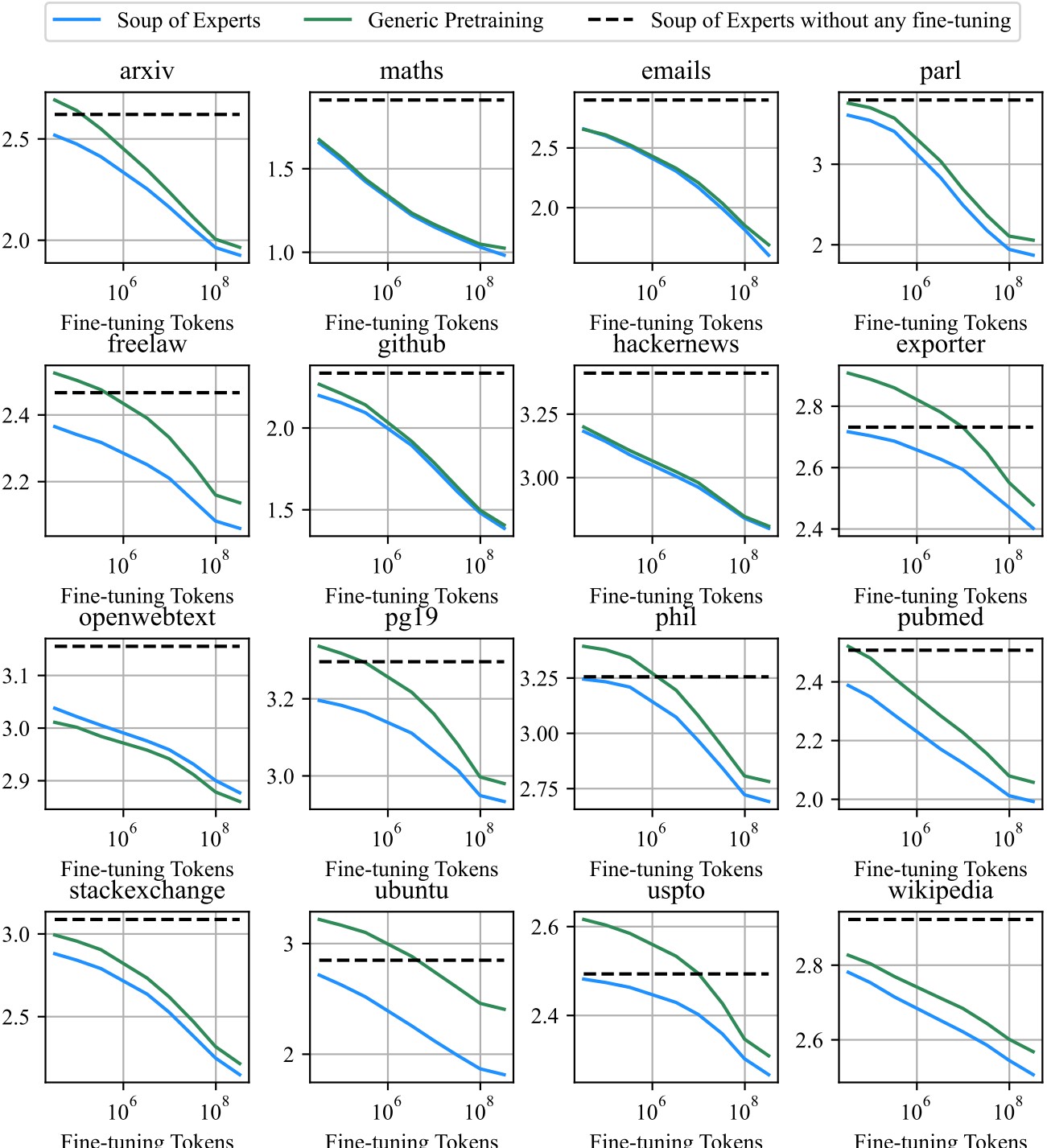

*Figure 12.* **Fine tuning results.** Detailed results from Figure 6 on the 16 PILE domains. The dashed horizontal line indicates the loss of the Soup-of-Experts without fine-tuning.

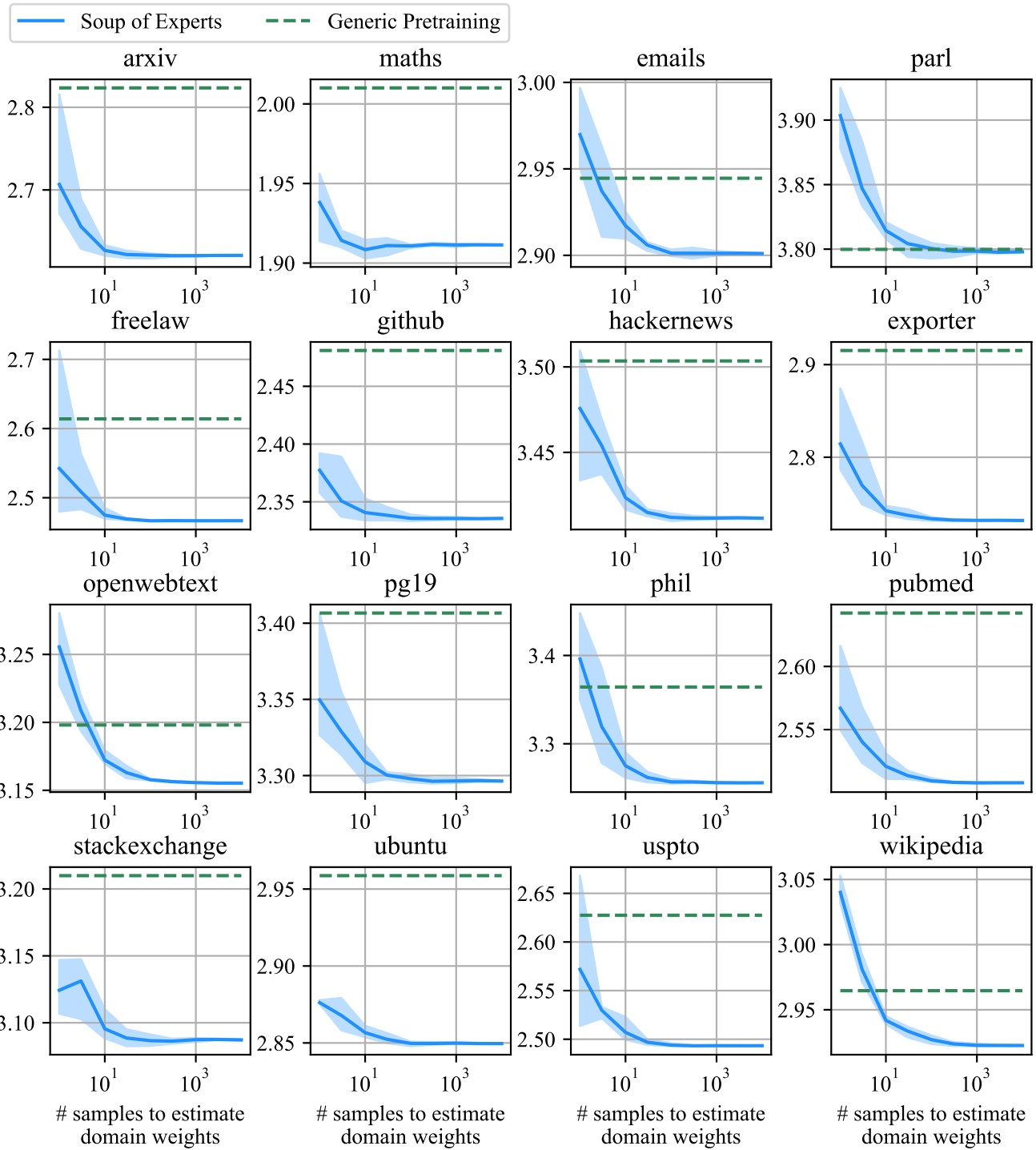

*Figure 13.* **Impact of the number of samples in the specific set on the instantiated Soup-of-Experts.** Detailed results from Figure 7, right, on the 16 PILE domains. The dashed horizontal line indicates the loss of the Soup-of-Experts without fine-tuning.

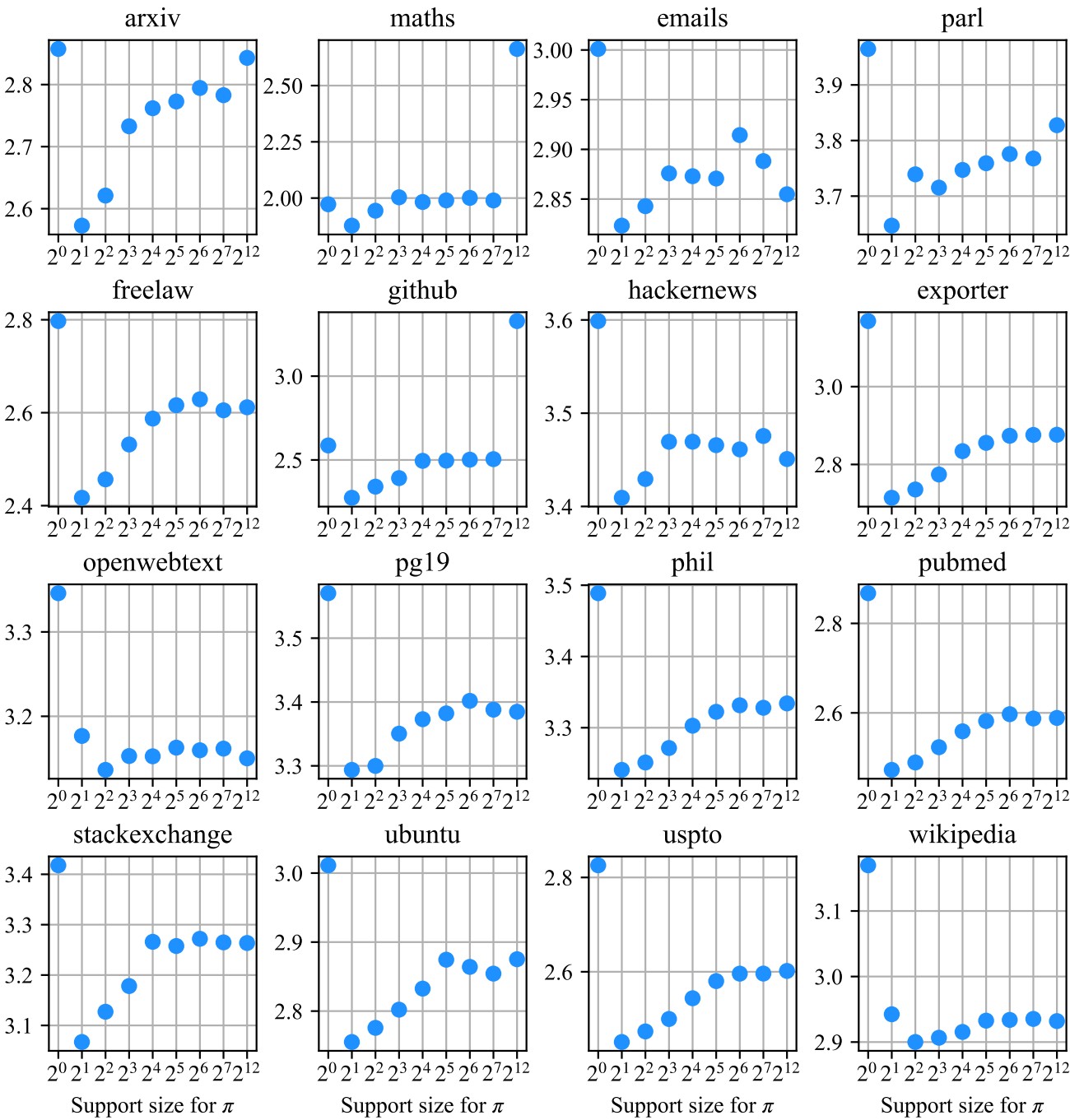

*Figure 14.* **Impact of the support size of the meta-distribution.** Detailed results from Figure 8, on the 16 PILE domains.

