# OpenReview forum: "Soup-of-Experts: Pretraining Specialist Models via Parameters Averaging"
_ICML.cc/2025/Conference — ICML 2025 poster_

### Official Review · Reviewer_pSxt · 2025-03-11

**Overall Recommendation:** 3

**Summary:**

The paper proposes a method for combining a set of pretrained models by learning linear coefficients that are used to merge corresponding layer weights, thereby producing domain-specific architectures. In essence, the approach learns weighting coefficients similar to a router in mixture-of-experts models that dynamically determine how to combine the expert parameters based on the input domain. Experimental results demonstrate that this strategy yields promising performance improvements on specialized tasks.

**Claims And Evidence:**

The paper makes several key claims, and overall, many of these are backed by experimental evidence. However, a few claims could benefit from additional substantiation.

**Essential References Not Discussed:**

The paper builds on ideas from model merging, MoE architectures, and importance sampling. However, some essential related works could have been discussed more explicitly to contextualize its contributions:
Several key papers that provide additional context to the proposed method were not discussed in the paper. These include:

- **[0] Merging Experts into One: Improving Computational Efficiency of Mixture of Experts:** This work explores methods to merge multiple experts into a single model to enhance computational efficiency.
- **[1] Mixture of LoRA Experts:** This paper investigates leveraging LoRA-based adaptations within a mixture-of-experts framework with learnable gate.
- **[2] Mixture of In-Context Experts Enhance LLMs’ Long Context Awareness:** This approach uses a mixture of experts to improve the long-context processing capabilities of large language models.
- **[3] REMOE: Fully Differentiable Mixture-of-Experts with ReLU Routing:** REMOE proposes a fully differentiable routing mechanism using ReLU, which addresses training stability and efficiency issues.
- **[4] Mixture-of-Experts with Expert Choice Routing:** This work introduces an alternative routing strategy where experts are chosen based on an expert-choice mechanism rather than traditional softmax gating.
- **[5] HyperTuning: Toward Adapting Large Language Models without Back-propagation:** HyperTuning explores rapid adaptation methods for large language models without relying on full back-propagation.
- **[6] HINT: Hypernetwork Instruction Tuning for Efficient Zero- & Few-Shot Generalisation:** HINT leverages hypernetwork-based techniques for instruction tuning, enabling efficient zero- and few-shot generalization.

Discussing these papers would enrich the context of the proposed method by comparing alternative strategies for expert merging, dynamic weight adaptation, and efficient model specialization.

**Experimental Designs Or Analyses:**

The experimental designs are well conceived and align with the goals of rapid model specialization under a size constraint. However, the authors focus solely on next-token prediction as the evaluation metric, which limits the depth of investigation into the proposed method’s broader performance on downstream tasks. Additionally, the method is not directly compared to other merging techniques. For instance, there is no experimental comparison with alternative merging approaches such as hypernetwork-based methods that dynamically select specialized weights even though the concept appears similar. This leaves an open question as to how the proposed method performs relative to other state-of-the-art merging strategies.

**Methods And Evaluation Criteria:**

The proposed methods and evaluation criteria in the paper are well-suited to the problem at hand. The central idea is to train a mixture-of-experts model on multiple datasets by optimizing a router that learns to weight the experts based on the input domain. At inference time, selected expert weights are linearly merged to quickly instantiate a specialist model tailored to a specific domain. The authors evaluate their approach on standard language modeling tasks using the next-token prediction loss

**Other Comments Or Suggestions:**

See above sections:

- In lines 047–051 (right column), the authors refer to “model soup” as a general term for model averaging in weight space. However, model soup is also a specific algorithm that starts with the best model, then averages with the second best if performance improves, uses that average as a new baseline for averaging with the third model, and so on—skipping any model that does not yield an improvement at a given step. It would be beneficial for the authors to clarify these differences when discussing the model soup concept.

- It would be valuable to compare the proposed method with existing expert merging approaches and evaluate its scalability on larger models.

- The captions are overly long, which makes them hard to read. It would improve clarity if detailed descriptions were moved to the discussion sections of the corresponding experiments instead of being included entirely in the captions.

**Other Strengths And Weaknesses:**

**Strengths:**

- The paper presents an effective strategy to reduce the challenge of generating domain expert models. By learning weights for merging experts into a single specialized model, it provides a computationally efficient alternative to training separate models for each domain.
- The proposed method is practically sound—it is scalable and designed for rapid model specialization, which is supported by comprehensive experimental results and ablation studies.

**Weaknesses:**

- The experimental evaluation is narrow, as it is limited to next-token prediction metrics. This focus may not fully capture the performance improvements on a broader range of downstream tasks.
- The authors do not adequately compare their approach to existing expert merging techniques (e.g., methods detailed in [0] and [1]). They present their method as a first without discussing how it outperforms other strategies, including hypernetwork-based approaches.
- The experiments are conducted on relatively small architectures, and given the computational resources available, it would be beneficial to evaluate the approach on larger models (e.g., at least 1B-parameter models) to better assess scalability.
- It is unclear whether the number of tasks used in the expert merging experiments aligns with those used in the Soup-of-Experts experiments, leading to ambiguity about the direct comparability of the approaches.

**Questions For Authors:**

1. How does your method compare to hypernetwork-based and other mixture-of-experts merging approaches in terms of performance, generalization, and in-distribution accuracy?

2. How does the individual performance of a model instantiated via the Soup-of-Experts framework compare to that of a task-specific fine-tuned model on in-distribution data?

3. Is there a universal guideline for selecting the prior used for the dataset weights sampler, especially when the eventual application domain of the model is unknown?

4. Given that the number of dataset-specific experts in the baseline (e.g., 64) is much lower than the number of domains used in the meta-soup (4096), doesn’t this disparity affect performance? While training 4096 individual models may be computationally prohibitive, the difference in exposure could naturally lead to a higher generalization capacity. Could you provide more explanation on this issue?

**Relation To Broader Scientific Literature:**

The paper synthesizes ideas from model merging, MoE architectures, and importance sampling into a unified framework. It extends prior findings by addressing the computational and scalability challenges inherent in training and serving specialized models, thereby contributing a novel method that is both theoretically grounded and practically relevant. However, the paper could benefit from a more detailed investigation into how its approach compares to hypernetwork-based methods which also enable rapid adaptation by dynamically selecting specialized weights and a more thorough discussion of alternative merging strategies.

**Theoretical Claims:**

The proofs are mathematically detailed and build upon well-established properties of Gaussian distributions and clustering. Overall, they appear to be correct under the stated assumptions.

---

> ### Author Rebuttal · Authors · 2025-03-31
>
> Dear reviewer,
>
> We thank you for your work reviewing our paper and for your comments. We are glad to read that "The experimental designs are well conceived", and we propose "a novel method that is both theoretically grounded and practically relevant".
>
> We now try to address your concerns.
>
> - "*the authors focus solely on next-token prediction as the evaluation metric.*"
>
> We have now trained bigger 350M models with 32 experts, and evaluated the corresponding models on MMLU, arc_easy and arc_challenge. The reported accuracy is here (https://anonymous.4open.science/r/rebuttal_soup-E8F9/accs.pdf); we see that the gains obtained in loss also transfer to accuracy.
>
> - "*the method is not directly compared to other merging techniques.*"
>
> See answer to rev.No4u where we compare SoEs to model merging.
>
> - "*The authors do not adequately compare their approach to existing expert merging techniques (e.g., methods detailed in [0] and [1])*"
>
> We will add the following discussion to the paper.
> [0] propose a variant of mixture of experts, where instead of mixing activations, they mix expert weights. However, the routing mechanism is still that of an MoE, based on input samples. On the contrary, our routing mechanism is based on input domains. For instance, one cannot use [0] to instantiate a small model, while it is one of the main features of SoEs.
>
> [1] propose a routing mechanism to merge a bank of Lora weights. Similarly to the previous work, this cannot instantiate a frozen model of small size in a flash, while SoEs can.
>
> - "*Several key papers that provide additional context to the proposed method were not discussed in the paper. These include*"
>
> Thanks for these useful references, we will discuss the most relevant in the paper.
>
> - "*The experiments are conducted on relatively small architectures, and given the computational resources available, it would be beneficial to evaluate the approach on larger models (e.g., at least 1B-parameter models) to better assess scalability.*"
>
> As discussed above, we have now trained 350M models with 32 experts.
>
> - "*It is unclear whether the number of tasks used in the expert merging experiments aligns with those used in the Soup-of-Experts experiments, leading to ambiguity about the direct comparability of the approaches.*"
>
> We have compared the two approaches (see above). We want to highlight that the both methods are not used in the same way; SoE is a pretraining technique, while merging is used on fine-tuned models.
>
> - "*model soup is also a specific algorithm [...] It would be beneficial for the authors to clarify these differences when discussing the model soup concept.*"
>
> Indeed, we will clarify in the text that we simply mean model averaging when speaking of model soups (i.e. we only consider *uniform* soups using the Model soups' paper terminology).
>
> - "*The captions are overly long*"
>
> We will simplify the captions.
>
> - "*How does your method compare to hypernetwork-based and other mixture-of-experts*"
>
> As explained above and in table1, SoEs and MoEs serve different purposes: MoEs cannot instantiate a small model, and they are not aware of the input domain distribution. We will clarify.
>
> - "*How does the individual performance of a model instantiated via the Soup-of-Experts framework compare to that of a task-specific fine-tuned model on in-distribution data?*"
>
> We believe this is the experiment in fig.5, where we compare SoEs to fine-tuned models.
>
> - "*Is there a universal guideline for selecting the prior used for the dataset weights sampler?*"
>
> In our experiments, we used ad-hoc priors that are completely blind to the downstream tasks that are eventually going to be addressed. We still obtain very good performances compared to regular pre-training. Incorporating knowledge about the downstream distribution of tasks is an exciting research direction.
>
> - "*Given that the number of dataset-specific experts in the baseline (e.g., 64) is much lower than the number of domains used in the meta-soup (4096), doesn’t this disparity affect performance?*"
>
> We also tried training models with 64 domains, but the performances where worse than with 4096 domains. The routing mechanism learns to combine those domains in a meaningful way that makes sense for the experts.
>
> - *"While training 4096 individual models may be computationally prohibitive, the difference in exposure could naturally lead to a higher generalization capacity. Could you provide more explanation on this issue?"*
>
> We believe that these models would be over-specific, and have no general knowledge at all. We conducted that experiment with 64 individual models in fig.4, and those models are all very poor generalists. Such a method would therefore perform poorly compared to SoE.
>
> We thank you again for your detailed review which help us improve the paper ! We hope that our comments have resolved your concerns.

---

### Official Review · Reviewer_wvpV · 2025-03-12

**Overall Recommendation:** 3

**Summary:**

This paper introduces Soup-of-Experts, a method that trains a group of expert models to construct a specialized model for a given target domain. The specialized model is obtained by linearly combining the expert models in parameter space. The architecture consists of a set of expert models, a shared model, and an MLP that generates blending weights based on input domain weights. During testing, the domain weights for each target domain are estimated either through an off-the-shelf method or a learning-based approach, leveraging training domain information for comparison. The proposed method demonstrates superior performance across various domains while maintaining a compact specialized model.

**Claims And Evidence:**

•  The explanation in Section 2.5 regarding the negligible overhead cost for training the group of experts is unclear. The authors state, “The forward pass through the experts also yields a negligible cost as long as the experts all fit in memory since it only requires adding several parameters.” However, this claim is confusing because the parameter space is duplicated n times, along with the saved feature maps. Additionally, both the forward computation and backpropagation scale linearly with the number of experts. Furthermore, it is unclear which “standard pre-training” is being referenced for comparison. The description in lines 306–309 suggests that the generic pretraining involves only a single model, making the comparison unclear.

•  The paper's primary motivation is to incorporate domain information during training to learn specialized experts. This approach suggests that capturing a diverse distribution of domains at the meta-level should be beneficial. However, the experimental results indicate that selecting only two domains (s = 2) and sampling uniformly from them yields the best performance. This seems counterintuitive, as s = 2 appears overly sparse given the total of 4096 training domains. Such a small sample may not adequately capture the complex relationships among domains. Moreover, iterating over all 4096 domains would require at least 2048 iterations. Yet, the authors claim that this approach can reduce both training cost and time, which seems unintuitive and requires further clarification.

•  Additionally, the model includes a shared branch, \mathcal{S}, which is common across all training domains and experts. Given the sparse domain sampling strategy, it is unclear whether this setup might lead to conflicts across different training iterations. Since \mathcal{S} is optimized toward the selected two domains at each iteration, substantial domain gaps between different domain pairs across iterations could lead to instability in training. The paper should address whether this issue affects model convergence and performance.

**Essential References Not Discussed:**

The references are thoroughly discussed.

**Experimental Designs Or Analyses:**

The experiment design and analysis are appropriate.

**Methods And Evaluation Criteria:**

Yes

**Other Comments Or Suggestions:**

N/A

**Other Strengths And Weaknesses:**

This paper delivers good results on how to reduce the resources and computation needed when the model is expected to be specialized to a particular domain.

The proposed method is also easy to understand and does not contain complex settings/modules. It will benefit the future research in the area, especially larges models become the focus,s but it is also important to focus more on the specialized scenarios.

**Questions For Authors:**

N/A

**Relation To Broader Scientific Literature:**

This paper is closely related to major learning paradigms in language models, including dense architectures and Mixture-of-Experts (MoE) structures. Additionally, the proposed method builds on the concept of model soup from previous literature, which averages models in parameter space to achieve a better tradeoff between in-distribution and out-of-distribution performance (Wortsman et al., 2022). The primary objective of this paper is to mitigate distribution shift and adapt the model more effectively to target domains.

**Theoretical Claims:**

There are no theoretical claims in this paper.

---

> ### Author Rebuttal · Authors · 2025-03-31
>
> Dear reviewer,
>
> We thank you for your review and for helping us improve the paper. We are happy to read that "The experiment design and analysis are appropriate.", that "this paper delivers good results", and that "the proposed method is easy to understand".
>
> We now try to address your concerns.
>
> - *" The explanation in Section 2.5 regarding the negligible overhead cost for training the group of experts is unclear. [...] Additionally, both the forward computation and backpropagation scale linearly with the number of experts."*
>
> This is an important remark, thanks ! We will greatly clarify this part with the following analysis: We compare training a model of size d with generic pre-training and training a Soup of Experts with n experts of size d. We let b be the mini-batch size.  The training loop of models consists of A) computing gradients and B) updating the parameters with Adam.
> For the generic model, the cost of A) scales roughly as $db$ [1], we let $C$ be the proportionality constant. This computes $\nabla\_\Theta \ell(\Theta, x)$, with a cost $C\cdot d\cdot b$.
> For the SoE, we need to compute the gradients w.r.t. the experts $E\_i$ and shared parameters $S$.  Let $\Theta = S + \sum \alpha\_i E\_i$ the merged model. Then, the chain rule gives the gradients $\nabla_S L = \nabla\_\Theta \ell(\Theta, x)$ and $\nabla\_{E\_i}L = \alpha_i \nabla\_\Theta \ell(\Theta, x)$. In other words, the gradients w.r.t. the experts are simply obtained by rescaling the gradient wrt the merged parameters, which gives $n\cdot b$ floating point operations. Hence, the cost of A) is $C\cdot d\cdot b + n\cdot b$ for the SoE. Crucially, **the cost of backprop through the SoE is only midly affected by n**, as the gradients of the experts are obtained trivially from those of the merged model; the cost is **not** $C\cdot b\cdot n\cdot d$, which would be the same cost as that of training a much larger $nd$ full model.
>
> For B, the generic pretraining needs to update the two adam EMA parameters, and then update the parameters.
> In total, the flops count is 14d. For the SoE, we need to use Adam for all the parameters, so the cost of B) is 14$n\cdot d$.
>
> Overall, one training iteration for generic pretraining costs $C_{gen} = Cdb + 14d$, while it is $C_{soe} = Cdb + 15nd$ for the SoE. Let us discard the cost of Adam for the generic pretraining, the relative increase is $C_{soe} / C_{gen} = 1 + \frac{15n}{Cb}$. Hence we see that it all depends on whether $n \gg Cb / 15$; the practical value of the constant $C$ depends on the architecture and other factors.
> In a large batch size $b$ setting, the added cost is negligible.
>
> We insist that in fig 4 and in https://anonymous.4open.science/r/rebuttal_soup-E8F9/reb_soe_time.pdf we report time as x-axis, where the additionnal (small) overhead is already factored in.
>
> - "*Furthermore, it is unclear which “standard pre-training” is being referenced for comparison"*
>
> We will clarify that standard pre-training trains one single model of size $d$, which in section 2.5. is compared to the cost of training one soup of experts with $n$ experts of the same size $d$.
>
> -"*the experimental results indicate that selecting only two domains (s = 2) and sampling uniformly from them yields the best performance. This seems counterintuitive[...]*"
>
> We want to clarify a possible misunderstanding: as explained in 2.4, we train using only $s$ active domains at a time, but these domains are sampled at random at each iteration. For instance, if we had only 3 domains and use $s=2$, sampling from $\pi$ might give these 4 samples: `[0, 0.2, 0.8], [0, 0.6, 0.4], [0.1, 0.9, 0], [0.7, 0, 0.3]`, where there are only 2 non-zero domain at each time, but their index change.
> Hence, this strategy allows us to explore the space of all pairs of domains, not just one fixed pair of domain. In practice, we use 256000 iterations, so the model has seen data coming from each domain multiple times. We hope that this clarifies the "counterintuitive" observation that the SoE accelerates training. We will add this explanation in the text to make it clearer.
>
> - "*the model includes a shared branch [...]. Given the sparse domain sampling strategy, it is unclear whether this setup might lead to conflicts across different training iterations.*"
>
> Indeed, as you mention, there is a tension between domains, and the shared branch has to accomodate between all domains, just like a standard LLM has to share its capacity across different domains. However, we did not observe any particular instabilities during training, the training hyperparameters (gradient clipping, adam parameters) are standard, and the curves in fig.4 are very smooth.
>
> We hope that our answers have addressed your concerns, and thank you again for your helpful review !
>
> [1]: Kaplan, Jared, et al. "Scaling laws for neural language models." arXiv preprint arXiv:2001.08361 (2020).

---

### Official Review · Reviewer_qpFf · 2025-03-17

**Overall Recommendation:** 3

**Summary:**

This paper studies the problem of training specialized models efficiently for a new domain. This is done by design from the pretraining stage by training a model in a style akin to mixture-of-experts. The data is sampled across domains in a mini-batch and each expert corresponding to each domain is updated, together with the general model that is always updated. This results in one model for each domain and a general model. For specialization, the data from the new domain is categorized using nearest neighbors based on semantic similarity into a distribution. This distribution is them used to perform a weighted merging of models via averaging to obtain the specialist model.

The paper uses a model of 110M parameters based on the GPT-2 architecture and pretrains on RedPajama2 128 different experts with data from 4096 domains that are obtained using k-means on semantic similarity. The loss of the model is shown to be worse on the generic data than that of the regular model. However, when using data from 16 domains of the Pile data set, fine-tuning the obtained specialized model shows better loss than the generic model.

The paper is overall well-written. The task setup that is being studied is very relevant and important.

In my opinion, the major drawback of the paper is that is does not show evaluation on downstream data sets and only shows the loss. Even the paper's abstract mentions that the downstream performance is the main motivation for this setup. Additionaly, perplexity would be another interesting metric to see.

Another important drawback of this approach is that in this experiment, we need to keep n=128 experts trained, which will require a lot of space. Further, in order to efficiently do the model updates, all these experts need to be stored in memory, which is prohibitive for large models.

## update after rebuttal

Thank you for your response.

The additional experiments are indeed valuable and show positive results. It would be good to learn more about them and also present multiple downstream evaluations on more than 3 data sets.

Regardless, I am willing to improve my overall assessment in light on these two new results.

**Claims And Evidence:**

Claims are not fully supported by the experiments as the downstream fine-tuning metrics and perplexity are lacking.

**Essential References Not Discussed:**

Branch-Train-Merge: Embarrassingly Parallel Training of Expert Language Models - Li et al 2022.
- this paper explores a similar setup

**Experimental Designs Or Analyses:**

The Pile data sets may not be very different to the data that is present in the RedPajama2 set, as that contains common crawl snapshots, and the pile data sets are most likely present inside these snapshots (e.g. wikipedia).

**Methods And Evaluation Criteria:**

The major weakness in my view of this paper is that is lacks evaluation on downstream fine-tuned models with tasks in the specalist domain distribution. Even though the loss looks better, further validation on fine-tuning is critical to prove the effectiveness of the approach.

Further, the models trained are very small for generative LLMs (110M parameters). I suggest running an experiment on a larger model. An alternative would be to try this on BERT/RoBERTa models. Another model trained on this setup will add additional strength and show more generality.

**Other Comments Or Suggestions:**

Page 5: sentence-bert (Devlin, 2018) is the wrong citation for sentence-bert. It would be good to know which specific model version was actually used for similarity.

**Other Strengths And Weaknesses:**

-

**Questions For Authors:**

-

**Relation To Broader Scientific Literature:**

See summary section

**Theoretical Claims:**

The main contribution of this paper is the new architecture for the pre-trained model. This is described well and in good amount of detail.

---

> ### Author Rebuttal · Authors · 2025-03-31
>
> Dear reviewer,
>
> We thank you for your review and for your insightful comments, which will help us improve the paper. We are happy to hear that "The paper is overall well-written" and that  "The task setup that is being studied is very relevant and important."
>
> We now address your concerns.
>
> - "*the models trained are very small for generative LLMs (110M parameters). I suggest running an experiment on a larger model.*"
>
> We have now trained larger 350M models, using 32 experts for the Soup-of-Experts. The training curves are available here: https://anonymous.4open.science/r/rebuttal_soup-E8F9/reb_soe_time.pdf. The behavior is essentially similar to that in fig.4 in the paper. Due to lack of resources, we could not train larger models.
>
> - *"the major drawback of the paper is that is does not show evaluation on downstream data sets and only shows the loss."*
>
> This is a good point. We have score the 350M models that we have trained on three datasets: MMLU, Arc-easy and Arc-challenge. We report the corresponding average accuracy as a function of time in this figure: https://anonymous.4open.science/r/rebuttal_soup-E8F9/accs.pdf
>
> We observe that the SoE consistently outperforms generic pre-training. We will add these results to the paper. We believe that this result indeed makes the paper more convincing.
>
> - *"Additionaly, perplexity would be another interesting metric to see."*
>
> In all our experiments, we train and report with the next-token prediction loss; the perplexity can therefore be obtained by taking the exp of our curves in fig. 4, 5, 6, 7. It will not change the ordering of methods.
>
> - *"Another important drawback of this approach is that in this experiment, we need to keep n=128 experts trained, which will require a lot of space. Further, in order to efficiently do the model updates, all these experts need to be stored in memory, which is prohibitive for large models."*
>
> Indeed, we will clarify in 2.5 that the method requires storing all the expert's weights, which might be problematic for large model sizes. However, in the case where memory is not an issue, Soup-of-Experts achieves significantly better performances than standard pretraining at a fixed computational budget, as reported in Fig. 4 and https://anonymous.4open.science/r/rebuttal_soup-E8F9/reb_soe_time.pdf.
>
> - "*Claims are not fully supported by the experiments as the downstream fine-tuning metrics and perplexity are lacking.*"
>
> See above, this is now fully fixed.
>
> - "*The major weakness in my view of this paper is that is lacks evaluation on downstream fine-tuned models with tasks in the specalist domain distribution. Even though the loss looks better, further validation on fine-tuning is critical to prove the effectiveness of the approach.*"
>
> As seen above, we have validated that these gains in loss also transfer to downstream tasks.
>
>
> - *"Another model trained on this setup will add additional strength and show more generality."*
>
> As explained above, we have now trained larger 350M models. We think that this indeed improves the generality of our paper. We acknowledge that we focus solely on the decoder-only LLM architecture, which is one of the most widely used architecture nowadays.
>
> - *"The Pile data sets may not be very different to the data that is present in the RedPajama2 set, as that contains common crawl snapshots, and the pile data sets are most likely present inside these snapshots (e.g. wikipedia)."*
>
> Indeed, this is a judicious and important remark. We used the Pile as it has a variety of downstream domains that overlap more (wikipedia) or less (dm_mathematics) with the pretraining domain, in order to cover a variety of practical cases. We will clarify this in the text.
>
>
> - "*Branch-Train-Merge: Embarrassingly Parallel Training of Expert Language Models - Li et al 2022.*"
>
> Thanks for the relevant reference that we missed, we will discuss it in the paper.
>
> - "*Page 5: sentence-bert (Devlin, 2018) is the wrong citation for sentence-bert. It would be good to know which specific model version was actually used for similarity.*"
>
> Thanks for spotting this ! Indeed, the model we use is a fine-tuned version of MPNET [1], obtained at https://huggingface.co/sentence-transformers/all-mpnet-base-v2. We will properly acknowledge this in the paper.
>
>
> We thank you again for your work reviewing our paper and for your comments which help us improve the paper. We hope that your concerns have been alleviated.
>
>
> [1] Song, Kaitao, et al. "Mpnet: Masked and permuted pre-training for language understanding." Advances in neural information processing systems 33 (2020): 16857-16867.

---

### Official Review · Reviewer_No4u · 2025-03-25

**Overall Recommendation:** 3

**Summary:**

The authors propose Soup-of-Experts, which can quickly create a specialized model for a given mixture of data domain weights at test time. Soup-of-Experts jointly pretrains and learns a function to compute weights for a linear combination of expert model weights for specialization.

## update after rebuttal
I updated the review by increasing the score from 2 to 3 after the authors addressed the majority of my concerns. I still see scaling as a drawback due to the large number of experts that need to be updated jointly. This also makes the method more inflexible since all of the experts need to be trained at once during pretraining.

**Claims And Evidence:**

The main claim, that soup-of-experts can be quickly used to obtain a pre-trained specialized model, seems somewhat well-supported by experiments comparing to other pretraining methods, as it performs slightly better than CRISP in the specialized loss while having much better performance in the generalized loss. It also has better specialization and similar general loss to pretraining a single similar-sized model.

**Essential References Not Discussed:**

There are a number of similar approaches in the domain adaptation realm in the fine-tuning setting which are not discussed, such as rewarded soups [1] and personalized soups [2]. It would be useful for the authors to provide comparisons to these approaches, since they only provide comparisons to pretraining-based approaches and it is not clear if pretraining results in gains over similar works combining fine-tuned models.

It would also be useful to discuss previous methods souping pretrained models without requiring joint pretraining from scratch, such as [3].

[1] Rame et. al. Rewarded soups: towards Pareto-optimal alignment by interpolating weights fine-tuned on diverse rewards
[2] Jang et. al. Personalized Soups: Personalized Large Language Model Alignment via Post-hoc Parameter Merging
[3] Imfeld et. al. Transformer Fusion with Optimal Transport.

**Experimental Designs Or Analyses:**

The experiments seem well-designed as a way to compare the models' general and domain adaptation capabilities. However, there are some additional baseline comparisons that would be useful (see Questions 1-2).

**Methods And Evaluation Criteria:**

The proposed method makes sense as a way to adapt model souping to pretraining. The benchmark datasets used have a variety of specialization domains and are useful for evaluation.

**Other Comments Or Suggestions:**

Typo in 2.5 section header

**Other Strengths And Weaknesses:**

Strengths:
1. The idea of jointly pretraining the experts and linear combination weights seems novel and interesting, compared to existing approaches that do one or the other
2. The experiments support the claim that this approach increases domain adaptation performance over existing pretraining-based methods

Weaknesses:
1. This approach seems very computationally expensive, since we first need a large number of domain experts which then need further pre-training to be adapted
2. The authors do not compare to or discuss approaches for souping fine-tuned models for domain adaptation.
3. Experiments are limited to small models (33M - 110M parameters)

**Questions For Authors:**

1. When is this method advantageous over using a single pretrained model of the same total number of parameters as is contained in the experts? What is the total training time comparison between this larger model when compared to training the experts and Soup-of-Experts?
2. How does this method compare to existing approaches combining fine-tuned models from multiple domains? Soup-of-Experts seems inherently less flexible since the experts must be pretrained from scratch rather than leveraging existing fine-tuned models.
3. Is this approach feasible for larger models? The large number of experts seems like a computational challenge if the models are scaled.

**Relation To Broader Scientific Literature:**

While model soups have been used previously with fine-tuned models and existing pretrained models, this paper instead focuses on test-time domain adaptation using linear souping by jointly pretraining expert models and a function on the domain weights.

**Theoretical Claims:**

There were no proofs

---

> ### Author Rebuttal · Authors · 2025-03-31
>
> Dear reviewer,
> Thanks a lot for your review and for your questions and remarks; they will help us improve the paper. We are please to see that you found that the main claim is well supported, that "the proposed method makes sense", and that the experiments are well designed.
>
> We now answer your questions and the points you have raised:
>
> - "*There are a number of similar approaches in the domain adaptation realm in the fine-tuning setting which are not discussed, such as rewarded soups [1] and personalized soups [2]. It would be useful for the authors to provide comparisons to these approaches,[...]*"
>
> Thanks for raising this point. While we already mentioned [1], we will discuss [2] and [3] in the paper. Thanks to your remark, we have implemented a comparison to standard model merging. To do so, we train both a SoE and a generic model during 64000 iterations, and for the SoE we use k=64 domains. Then, for each k domain, we fine-tune the generic model for T steps on that domain, yielding models $\theta_1,\dots \theta_k$ which are specialists for each domain.
>
> We then take $d$ domains among the $k$ at random, and either a) instantiate the SoE with weights uniform over the $d$ domains, 0 otherwise, or b) merge the specialized models $\theta_i$ where i covers the $d$ domains. We use a linear combination of models to merge. We then report the average loss of the corresponding model a) or b) on the $d$ chosen domains. For the model merging, we always pick the number of fine-tuning steps T in [1000, 2000, ..., 10000] that yields the smallest loss for each individual experiment. The results are here: https://anonymous.4open.science/r/rebuttal_soup-E8F9/soe_vs_merge.pdf
>
>  When there is only $1$ domain, fine tuning is the best method, since it has trained a specialist. As soon as we need to merge >= 4 specialists, Soup-of-Experts become advantageous. Importantly, the SoE did not have to be trained on each domain individually: in total, it has done 64000 iterations, while the model merging required 64000 + 64 * 10000 steps, so about 10 times more compute ! We will add this insightful experiment to the paper.
>
> - "*This approach seems very computationally expensive, since we first need a large number of domain experts which then need further pre-training to be adapted*"
>
> We will clarify the text, the computational cost of the method is not high as long as we have enough memory to train the models. For instance, the x-axis in fig4 is the number of hours to train the models, the fact that the SoE is better means that it reaches better losses in less time; it is therefore not more expensive.
>
> - "*The authors do not compare to or discuss approaches for souping fine-tuned models for domain adaptation.*"
>
> See above.
>
> - "*Experiments are limited to small models (33M - 110M parameters)*"
>
> We have now trained a 350M model with 32 experts for 33B tokens,, see https://anonymous.4open.science/r/rebuttal_soup-E8F9/reb_soe_time.pdf . The method scales: it still is about twice as fast as generic pretraining to reach a certain level of specialized loss. We will add this to the paper.
>
> - *"When is this method advantageous over using a single pretrained model of the same total number of parameters as is contained in the experts? What is the total training time comparison between this larger model when compared to training the experts and Soup-of-Experts?"*
>
> This is an interesting question. To give a concrete order of magnitude, in the experiments on the paper we use 110M models with 128 experts, yielding in total 13B parameters. There are many differences between the SoE and a 13B model. First, the SoE will most likely have a much higher perplexity, since it instantiates a model 100 times smaller. However, as a trade-off, it will have a much higher throughput since it is a 100 times smaller model, and because of that the training costs are also orders of magnitude different. For instance, training a 13B model may take weeks with 128 GPUs (see table 2 in [3]), while our training takes ~1 day with 8 gpus. Overall, these models are very different. We will clarify this in the text.
>
> - *"How does this method compare to existing approaches combining fine-tuned models from multiple domains?"*
>
> See response above. Indeed, Soup-of-Experts is a pretraining method, it should be seen as complementary to usual model merging.
>
> - "*Is this approach feasible for larger models? The large number of experts seems like a computational challenge if the models are scaled.*"
>
> As said above, we could train larger 350M models on one node of 8 Gpus. We could train larger models with more computational ressources, but this is beyond our constraints.
>
> We hope that we have convincingly answered your questions, and we thank you again for your review !
>
> [3]: Touvron, Hugo, et al. "Llama 2: Open foundation and fine-tuned chat models." arXiv preprint arXiv:2307.09288 (2023).

---

### Official Review · Reviewer_4deQ · 2025-03-28

**Overall Recommendation:** 3

**Summary:**

The authors introduce "Soup-of-Experts" a method for pretraining language models so that they can quickly instantiate small, specialized models for different domain distributions. The architecture consists of a shared base model, multiple expert parameter sets, and a small learned routing function (an MLP) that maps domain weights (i.e., mixtures of training domains) to a set of coefficients used to linearly combine the experts at inference. Since each specialist is simply a linear combination of trained parameters (plus the shared parameters), it can be instantiated in a single forward pass of the routing MLP, with no further training or fine-tuning needed. Experiments on several language modeling tasks (drawn from RedPajama for pretraining and from the Pile for specialization) show that the proposed approach can produce strong specialist models that significantly outperform a single generic model while only slightly sacrificing performance on a broad domain. Soup-of-Experts also outperforms other baselines (Domain Experts, CRISP) in terms of specialization performance and computational efficiency.

**Claims And Evidence:**

I consider the claims to be supported by sound evidence. Details are provided below:

Claim (1) - SoE can quickly produce specialized models targeted at specific domains by simply merging expert parameters with learned coefficients (no fine-tuning needed) >> The authors show that for 16 domains from Pile, SoE achieves lower perplexity/loss than a comparable single, generically pre-trained model.

Claim (2) - SoE is scalable >> The authors compare SoE to two baselines that train separate models for each domain. These baselines incur linear overhead (one pretraining done per domain), whilst SoE has a single pretraining phase for all possible domain mixture, then re-combines parameters at specialization.

Claim (3) - SoE almost preserved general performance >> Figure 4 confirms that SoE's overall perplexity on the broad (generic) dataset is only slightly worse than that of a purely generic model, suggesting that SoE retains broad-domain knowledge while also being adaptable.

**Essential References Not Discussed:**

The references appear sufficient to contextualize the core idea, but I did not carefully check for other similar references.

**Experimental Designs Or Analyses:**

Experiments are methodologically consistent (fair), meaning that each approach is trained with similar computational budgets (in tokens) and is then evaluated under the same metrics (next-token prediction loss). The authors report both the specialized loss and the generic training loss to show the trade-off between specialization and generality.

**Methods And Evaluation Criteria:**

From a reviewer’s standpoint, the methods and evaluation criteria are well aligned with the stated goal of obtaining strong specialized models from a single pretraining run. The choice of next-token prediction loss to measure performance is standard for language modeling tasks.

**Other Comments Or Suggestions:**

It would have been helpful to explicitly clarify how sensitive the results are to different embedding methods used for domain clustering.

Consider providing additional runtime benchmarks or memory usage comparisons explicitly against baselines.

**Other Strengths And Weaknesses:**

Strengths:
- Clear motivation and problem formulation, well-written submission.
- Novel combination of existing ideas (model merging + MoE-inspired expert routing).
- Empirical validation across datasets.
- Extensive ablation studies providing insights into method robustness (in the supplementary).
- Practical significance due to low computational overhead at specialization time.

Weaknesses:
- Low-rank experts did not yield expected improvements; there is a need for further investigation.
- Potential scalability challenges when significantly increasing the number of experts beyond tested scenarios.
- Limited theoretical analysis or no formal guarantees regarding stability.

**Questions For Authors:**

It would be helpful if the authors clarified the effect of the MLP capacity on final performance. Does a larger routing network yield more fine-grained domain combinations?

Also, I am curious to know how sensitive is the method's performance to the choice of embedding method used during the clustering of pretraining domains?

**Relation To Broader Scientific Literature:**

The authors position SoE relative to large language models (generic but expensive) and specialized smaller models (efficient but less general). They frame SoE as a novel extension of “model soup” ideas and parametric interpolation methods but used during pretraining with learned mapping from domain weights to combination coefficients.

**Theoretical Claims:**

* The paper does not provide a formal proof of correctness beyond standard machine-learning assumptions about parameter interpolation, so there are no complex proofs to verify.

* The theoretical argument is mostly intuitive, referencing earlier works on “model soups” / model merging and showing that experts with a shared initialization can be combined (especially when they are simultaneously trained).

---

> ### Author Rebuttal · Authors · 2025-03-31
>
> Dear reviewer,
>
> First, we thank you for your work, and for your remarks that will help us improve our work. We are happy to hear that *"the claims to be supported by sound evidence"*, that *"the methods and evaluation criteria are well aligned with the stated goal"*, and that *"Experiments are methodologically consistent"*.
>
> We now address your concerns.
>
> - *"Low-rank experts did not yield expected improvements; there is a need for further investigation."*
>
> Indeed, we could not make this idea work, and honestly report it in the paper. We think making it work would enable another avenue to scale soup-of-experts, even though we could scale them without Lora (see https://anonymous.4open.science/r/rebuttal_soup-E8F9/reb_soe_time.pdf where we train 350M models, with 32 experts for the SoE).
>
> - *"Potential scalability challenges when significantly increasing the number of experts beyond tested scenarios."*
>
> We need to select a tradeoff between the number of experts and model size, at a fixed memory size. We have been able to train 350M models with 32 experts, see https://anonymous.4open.science/r/rebuttal_soup-E8F9/reb_soe_time.pdf
>
> - *"Limited theoretical analysis or no formal guarantees regarding stability."*
>
> Indeed, we do not conduct theoretical analysis, as we are unsure what type of results we would want to prove; note that papers in a similar flavor like [1,2,3] also do conduct any theoretical analysis
>
> - *"It would have been helpful to explicitly clarify how sensitive the results are to different embedding methods used for domain clustering."* and *"Also, I am curious to know how sensitive is the method's performance to the choice of embedding method used during the clustering of pretraining domains?"*
>
> Thanks for the suggestion. We will clarify in the text that we use this type of embeddings since it is the one which offers the best performance are reported in [4], fig.5. Due to lack of time in the rebuttal period, we could not obtain training runs using different embeddings, but we will make sure to add this ablation to the final version.
>
> - *"Consider providing additional runtime benchmarks or memory usage comparisons explicitly against baselines."*
>
>
> We note that fig.4 gives a time comparison between methods, just like the newly added larger scale experiment https://anonymous.4open.science/r/rebuttal_soup-E8F9/reb_soe_time.pdf .
> The baseline and Soup-of-Experts training use the same hardware, that is one node of 8 A100 GPUs, and both have a GPU utilization of over 95%. We wil include these datapoints in the final paper.
>
>
> - *"It would be helpful if the authors clarified the effect of the MLP capacity on final performance. Does a larger routing network yield more fine-grained domain combinations?"*
>
> We have run an ablation where we change the width of the routing network. The MLP has input dimension 4096, output dimension 128, and we either take a hidden dimension of 4 * input_dimension or 4 * output_dimension. We could not see any difference between those two choices; the training curves looked similar. We will make sure to add a more thorough ablation in the final version of the paper.
>
>
> We hope that our response clarifies the questions you have raised, and we thank you again for your thorough review.
>
> [1]: Ha, David, Andrew Dai, and Quoc V. Le. "Hypernetworks." arXiv preprint arXiv:1609.09106 (2016).
>
> [2]: Krajewski, Jakub, et al. "Scaling laws for fine-grained mixture of experts." arXiv preprint arXiv:2402.07871 (2024).
>
> [3]: Wortsman, Mitchell, et al. "Model soups: averaging weights of multiple fine-tuned models improves accuracy without increasing inference time." International conference on machine learning. PMLR, 2022.
>
> [4]:Grangier, David, et al. "Task-adaptive pretrained language models via clustered-importance sampling." arXiv preprint arXiv:2410.03735 (2024).

---

### Decision · Program_Chairs · 2025-05-01

**Decision:**

Accept (poster)

**Comment:**

**Summary of the Paper:**
The paper introduces "Soup-of-Experts" (SoE), a novel architecture designed to quickly instantiate small, specialized models for different domain distributions. The architecture consists of a shared base model, multiple expert parameter sets, and a small learned routing function that maps domain weights to a set of coefficients used to linearly combine the experts at inference. The method is evaluated on several language modeling tasks, demonstrating its ability to produce strong specialist models that outperform a single generic model while maintaining computational efficiency.

**Strengths and Weaknesses Highlighted by Reviewers:**
**Strengths:**
  - Clear motivation and problem formulation.
  - Novel combination of existing ideas (model merging and MoE-inspired expert routing).
  - Empirical validation across multiple datasets.
  - Extensive ablation studies providing insights into method robustness.
  - Practical significance due to low computational overhead at specialization time.

**Weaknesses:**
  - Limited theoretical analysis or formal guarantees regarding stability.
  - Potential scalability challenges when significantly increasing the number of experts.
  - Initial experiments were limited to relatively small models (33M - 110M parameters), later extended to 350m models

**Rebuttal and Discussion with Reviewers:**
The authors provided additional experiments with larger models (350M parameters) and demonstrated that the gains in loss also transfer to downstream tasks. They addressed concerns about scalability by showing that the method scales well with larger models and provided comparisons to model merging techniques. The authors clarified the computational cost and the impact of the routing network's capacity on performance. Reviewers acknowledged the authors' responses and updated their assessments positively.

**Recommendation for (Weak) Acceptance:**
We recommend a weak acceptance for this paper. The proposed framework, particularly its joint pretraining the experts and linear combination weights, is a small but novel departure from existing mixture of experts / expert souping methods. The authors have provided substantial empirical evidence to support their claims, and the additional experiments conducted during the rebuttal phase have strengthened the paper. While there are some concerns about scalability and theoretical analysis, the approach has sufficent potential upside to make it an interesting proof-of-point contribution to the conference program.